# WEBARBITER: A PRINCIPLE-GUIDED REASONING PROCESS REWARD MODEL FOR WEB AGENTS

**Yao Zhang**[1,3*†], **Shijie Tang**[1*], **Zeyu Li**[2], **Zhen Han**[1†], **Volker Tresp**[1,3]

[1]LMU Munich  [2]Technical University of Munich  [3]Munich Center for Machine Learning (MCML)

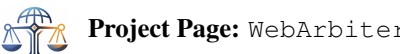

**Project Page:** `WebArbiter`

## ABSTRACT

Web agents hold great potential for automating complex computer tasks, yet their interactions involve long-horizon, sequential decision-making with irreversible actions. In such settings, outcome-based supervision is sparse and delayed, often rewarding incorrect trajectories and failing to support inference-time scaling. This motivates the use of Process Reward Models (WebPRMs) for web navigation, but existing approaches remain limited: scalar WebPRMs collapse progress into coarse, weakly grounded signals, while checklist-based WebPRMs rely on brittle template matching that fails under layout or semantic changes and often mislabels superficially correct actions as successful, providing little insight or interpretability. To address these challenges, we introduce **WebArbiter**, a reasoning-first, principle-inducing WebPRM that formulates reward modeling as text generation, producing structured justifications that conclude with a preference verdict and identify the action most conducive to task completion under the current context. Training follows a two-stage pipeline: reasoning distillation equips the model with coherent principle-guided reasoning, and reinforcement learning corrects teacher biases by directly aligning verdicts with correctness, enabling stronger generalization. To support systematic evaluation, we release WEBPRMBENCH, a comprehensive benchmark spanning four diverse web environments with rich tasks and high-quality preference annotations. On WEBPRMBENCH, WebArbiter-7B outperforms the strongest baseline, GPT-5, by 9.1 points. In reward-guided trajectory search on WebArena-Lite, it surpasses the best prior WebPRM by up to 6.4 points, underscoring its robustness and practical value in complex web tasks.

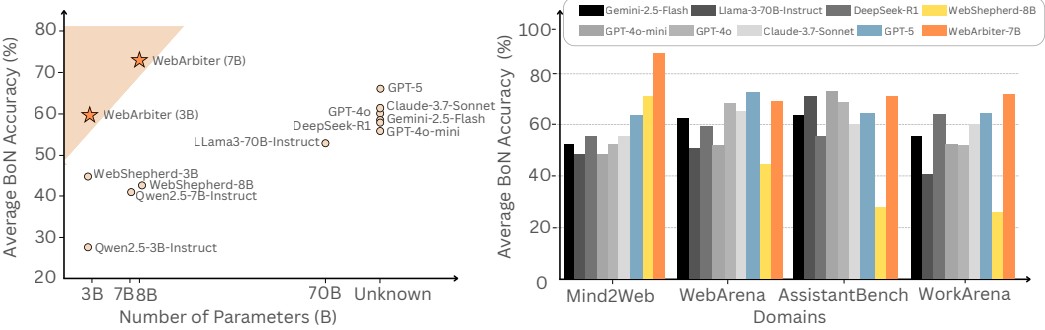

Figure 1: Performance comparison on WEBPRMBENCH. **Left:** *Average Best-of-N Acc* vs. model size, showing superior efficiency despite smaller scale. **Right:** Domain-wise *Avg BoN Acc*, where WebArbiter achieves the best results across all environments, confirming robustness and scalability.

# 1 INTRODUCTION

Large Language Models (LLMs) (Achiam et al., 2023; Guo et al., 2025a) have demonstrated impressive capabilities in planning (Huang et al., 2024; Zhang et al., 2025a), decision-making (Li et al., 2024), and complex task execution (Xi et al., 2024; Zhang et al., 2025b). Extending these abilities with browser access enables LLM agents to perform complex web tasks similar to humans (OpenAI, 2025a; Anthropic, 2024a; Adept, 2022). However, web interactions involve long horizons, multi-step decisions, and actions that can be irreversible. For example, submitting an incorrect form may not be recoverable. This requires agents to make reliable decisions throughout the interaction process, rather than relying solely on final outcomes. Traditional Outcome Reward Models (ORMs) are ill-suited: they provide only sparse and delayed feedback, may misclassify incorrect trajectories as successes, and cannot guide inference-time strategies, such as reward-guided search.

Recent studies on web agents (Zhang et al., 2025b; Koh et al., 2025) have introduced step-level rewards using LLM-as-judge. While such supervision can be useful, LLM-as-judge suffers from high cost, limited scalability, and susceptibility to hallucination, often rewarding fluent but incorrect actions. This motivates the development of dedicated Process Reward Models (WebPRMs) for web tasks. Existing WebPRMs largely fall into two categories: scalar WebPRM (Miao et al., 2025), which collapse progress into coarse scores with little interpretability or weak grounding; and generative WebPRM (Chae et al., 2025), which rely on checklists that are brittle under dynamic layouts and state-dependent action semantics. Moreover, lacking explicit reasoning, generative WebPRMs remain vulnerable to surface correlations and sensitive to page changes. These limitations highlight the need for a reasoning-first WebPRM that can verify progress, resist superficial biases, and provide interpretable chains for diagnosing errors.

To this end, we propose **WebArbiter**, a reasoning-first, principle-inducing WebPRM. It formulates process reward modeling as text generation: given task context and candidate actions with their reasoning traces, the model produces a structured justification that concludes with a preference verdict, identifying the action most conducive to task completion. Unlike scalar scores or checklist-based methods tied to fixed templates, WebArbiter dynamically derives principles from user intent and the current state, incorporates them into reasoning chains that verify whether an action advances task completion. Training follows a two-stage pipeline: reasoning distillation equips the model with coherent principle-guided reasoning, and reinforcement learning (RL) corrects teacher biases and aligns verdicts with correctness. This design transforms reward signals from shallow correlations into auditable analyses, making judgments robust to environment and page variations, resistant to spurious cues, and accurate in credit assignment.

To advance the evaluation of WebPRMs, we introduce WEBPRMBENCH, the first comprehensive evaluation benchmark spanning diverse environments dedicated to WebPRMs. It provides 1,150 step-level preference instances, each consisting of one correct action and four rejected alternatives, collected across 4 web environments: AssistantBench (Yoran et al., 2024), Mind2Web (Deng et al., 2023), WorkArena (Drouin et al., 2024; Boisvert et al., 2025), and WebArena (Zhou et al., 2023). The tasks span everyday activities such as online shopping and forum posting, as well as enterprise scenarios like updating schedules in IT management platforms. By combining scale, diversity, and fine-grained supervision, WEBPRMBENCH establishes a unified standard for systematic evaluation of WebPRMs, with *Pairwise* and *Best-of-N (BoN) Accuracy* as the primary metrics.

As shown in Fig. 1, experiments on WEBPRMBENCH show that WebArbiter achieves the highest *Avg. BoN Acc* among all evaluated models, outperforming the strongest proprietary LLM baseline, GPT-5, by 9.1 points, and consistently surpassing the previous SOTA WebPRM, WebShepherd, across all environments. Beyond static evaluation, WebArbiter also proves effective in practice: in reward-guided trajectory search on WebArena-Lite (Liu et al., 2024b), it delivers substantial gains, surpassing WebShepherd by up to 6.4 points, further demonstrating robustness in realistic interaction settings.

The key contributions of this work are:

1. We propose WebArbiter, a reasoning-first, principle-inducing PRM trained with reasoning distillation and RL, providing auditable reasoning chains and correctness-aligned signals.

---

*Equal contribution.
†Corresponding authors.

2. We release WEBPRMBENCH, the first comprehensive evaluation benchmark to provide systematic WebPRM evaluation across 4 web environments, using *Pairwise* and *Best-of-N (BoN) Accuracy* as standard metrics.

3. We show that WebArbiter achieves SOTA performance on WEBPRMBENCH, surpassing both proprietary LLMs and the previous SOTA WebPRM. In reward-guided trajectory search on WebArena-Lite, WebArbiter delivers gains of up to 6.4 points in success rate over the best prior WebPRM.

4. We analyze the effects of different training components through systematic ablations, showing that cold-start RL alone is unstable across environments, whereas reasoning distillation and explicit principles are essential for stable and transferable progress-aware judgments, with RL primarily acting as an amplifier.

## 2 RELATED WORK

### 2.1 LLM-BASED AUTONOMOUS WEB AGENTS

LLM advances have enabled browser-operating agents (Kim et al., 2024; Sun et al., 2024; Prasad et al., 2023; Fu et al., 2024; Ma et al., 2023; Zheng et al., 2023; Tao et al., 2023). One line distills environment-specific state–action pairs from demonstrations, strong on seen states yet brittle on novel ones, with SteP as a leading example on WebArena (Sodhi et al., 2024; Zhou et al., 2023). A second line pursues open-ended exploration via reflexive evaluation and search (Pan et al., 2024; Shinn et al., 2024; Koh et al., 2024; Zhang et al., 2025b). A third direction applies RL (Qi et al., 2025; Wei et al., 2025), yet real sites provide sparse and delayed signals, which makes value learning unstable without dense step feedback. Therefore, WebAgents require a process-level judge that assesses progress step by step and supplies auditable signals for search and planning.

### 2.2 REWARD MODELS IN REASONING AND WEB TASKS

RMs fall into two families. Scalar RMs attach a single numeric score to a response and use either absolute or discriminative schemes for evaluation (Uesato et al., 2022; Ouyang et al., 2022; Liu et al., 2024a; 2025; Park et al., 2024; Wang et al., 2024a; 2023b; 2024b). Generative RMs instead produce natural–language feedback from which rewards are extracted, aligning with LLM-as-Judge and supporting both single-instance evaluation and multi-response comparison; they show promising scalability but raise reliability concerns due to bias and hallucination (Lightman et al., 2023; Wang et al., 2023a; Zhang et al., 2025d; Wu et al., 2024; Ye et al., 2025; Zhang et al., 2024; 2025c). Building on these, Reasoning RMs cast judging as a deliberate process: they first generate an explicit, context-grounded chain of principle and analysis, then issue a single preference verdict, yielding adaptive test-time compute, stronger grounding, and interpretable feedback (Chen et al., 2025; Guo et al., 2025b; Mahan et al., 2024). In web agents, action rewards have been derived by the following methods: LLM-as-Judge (Zhang et al., 2025b; Koh et al., 2025), slow and unstable during search; scalar scoring (Miao et al., 2025), which collapses progress into coarse values with little interpretability and weak grounding; and checklist-driven generative feedback (Chae et al., 2025), whose external templates are brittle under layout and semantic drift and prone to surface correlations. These limitations motivate a reasoning-first approach that turns rewards from shallow correlations into auditable analyses. WebArbiter produces structured justifications with a single preference verdict, induces principles from the current instruction and state, and is trained by reasoning distillation followed by RL, so that judgments remain robust to environment variations, resist spurious cues, and provide accurate credit assignment while supporting inference-time scaling.

## 3 METHODOLOGY

In this section, we present the design of WebArbiter. Fig. 2 provides an overview of the WebArbiter framework, including the two-stage training pipeline and the inference-time principle-guided decision process. We begin by framing web navigation as a Partially Observable Markov Decision Process (POMDP) in §3.1, then describe how we construct a pairwise-preference dataset for training in §3.2. We introduce the training pipeline of WebArbiter model in §3.3. For clarity, we summarize all notations in Appendix A.

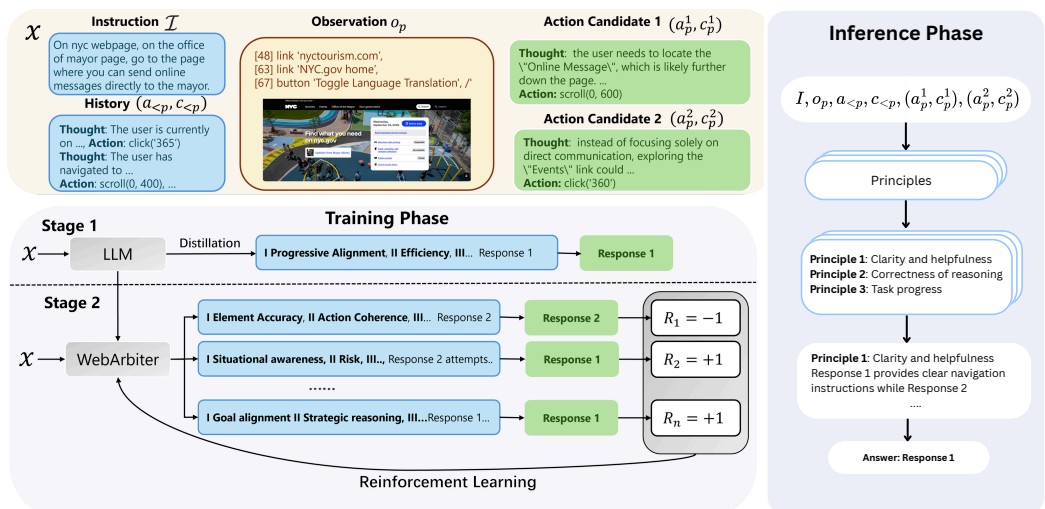

Figure 2: Overview of WebArbiter. Given an instruction $\mathcal{I}$, current observation $o_p$, and history $(a_{<p}, c_{<p})$, the model compares candidate actions $(a_p^1, c_p^1)$ and $(a_p^2, c_p^2)$. In **Stage 1**, principle-guided reasoning traces are distilled from a stronger teacher LLM. In **Stage 2**, WebArbiter is trained with RL using verifiable rewards $R \in \{-1, +1\}$, producing structured justifications and a final verdict. During inference, the model induces principles (e.g., clarity, correctness, progress) from $(\mathcal{I}, o_p, a_{<p}, c_{<p}, (a_p^1, c_p^1), (a_p^2, c_p^2))$, applies them to candidate actions, and outputs an auditable judgment identifying the action that best advances task completion.

## 3.1 BACKGROUND

We formalize web navigation as a POMDP. The environment $\mathcal{E}$ is defined by a state space $\mathcal{S}$, an action space $\mathcal{A}$, and an observation space $\mathcal{O}$. $T : \mathcal{S} \times \mathcal{A} \to \mathcal{S}$ denotes the state transition function. At step $p$, the agent receives a partial observation $o_p \in \mathcal{O}$, executes $a_p \in \mathcal{A}$, and transitions to $s_{p+1} = T(s_p, a_p)$ with a new observation $o_{p+1}$. Following WebArena (Zhou et al., 2024), we represent observations using accessibility trees, i.e., text-only encodings of visible interactive elements and their attributes. Given a task instruction $\mathcal{I}$ and the initial state $s_0 \in \mathcal{S}$, the agent aims to generate a trajectory $\tau = (a_1, \ldots, a_P)$ that completes the task. Here $P$ is the trajectory length and $a_p \in \mathcal{A}$ denotes the action at step $p$. The task evaluator determines whether the task is completed based on the final state.

## 3.2 TRAINING DATASET CONSTRUCTION

We build on the WEBPRM COLLECTION (Chae et al., 2025) for training WebArbiter. Each instance consists of an instruction $\mathcal{I}$, a sequence of observations $O = (o_1, \ldots, o_P)$, and expert-annotated trajectories. Specifically, the dataset provides a set of positive actions $A^+ = (a_1^+, \ldots, a_P^+)$ taken from expert demonstrations and negative actions $A^- = (a_1^-, \ldots, a_P^-)$ obtained from rejected trajectories. We convert these into pairwise preference samples where each candidate action is paired with its reasoning trace, yielding the preference dataset $\mathcal{D}_{\text{Train}}$ used for WebArbiter training.

## 3.3 WEBARBITER: A PRINCIPLE-INDUCING REASONING PROCESS REWARD MODEL

WebArbiter is built on a Transformer-decoder backbone and formulates process reward modeling as a text generation task. At each state, it evaluates candidate actions $\{(a_p^q, c_p^q)\}_{q=1}^{Q}$, where each action $a_p^q$ is paired with a reasoning trace $c_p^q$ explaining why the agent generated this action. Given task instruction $\mathcal{I}$, observation $o_p$, and history $(a_{<p}, c_{<p})$, the model autoregressively generates a structured justification $j = (j_1, \ldots, j_L)$ of length $L$ that concludes with a preference verdict $\hat{y}$ selecting the most appropriate action. The historical traces are $c_{<p} = \{c_1, \ldots, c_{p-1}\}$, i.e., the per-action reasoning traces for previously executed actions. A concrete training example is provided in Appendix B. While our experiments instantiate this framework in the standard pairwise preference setting, the design is general and extends naturally to multi-candidate settings.

Unlike the scalar WebPRM (Miao et al., 2025) that collapses progress into opaque scores or the checklist-based WebPRM (Chae et al., 2025), WebArbiter is a reasoning-first, principle-inducing WebPRM: it derives principles from user intent and the current state, integrates them into reasoning chains that explicitly assess whether each candidate action advances task completion. This design moves reward signals beyond shallow correlations toward auditable analyses, yielding judgments that are robust to environment changes, resistant to spurious cues, and precise in credit assignment.

Formally, the preference dataset is defined as

$$\mathcal{D}_{\text{Train}} = \{(\mathcal{I}^{(i)}, o_p^{(i)}, a_{<p}^{(i)}, c_{<p}^{(i)}, (a_p^{1(i)}, c_p^{1(i)}), (a_p^{2(i)}, c_p^{2(i)}), y^{(i)})\}_{i=1}^M, \tag{1}$$

where $y \in \{a_p^1, a_p^2\}$ denotes the preferred action. For notational simplicity, let

$$x = (\mathcal{I}, o_p, a_{<p}, c_{<p}, (a_p^1, c_p^1), (a_p^2, c_p^2)). \tag{2}$$

WebArbiter $\pi_\theta$ is parameterized by $\theta$ and models the justification $j$ autoregressively:

$$\pi_\theta(j \mid x) = \prod_{l=1}^L \pi_\theta(j_l \mid x, j_{<l}). \tag{3}$$

### 3.3.1 TRAINING OVERVIEW

The overall training objective is to maximize the likelihood that the predicted preference matches the ground truth:

$$\max_{\pi_\theta} \quad \mathbb{E}_{(x,y)\sim\mathcal{D}_{\text{Train}}, \ \hat{y}\sim\pi_\theta(j|x)} \left[ \mathbb{1}(\hat{y} = y) \right]. \tag{4}$$

Training proceeds in two stages. First, reasoning distillation, described in §3.3.2, equips the model to generate coherent, principle-guided justifications, promoting judgments grounded in explicit reasoning rather than surface correlations, a property validated by ablation studies in §5.1.3. Concretely, we sample $K$ examples from $\mathcal{D}_{\text{Train}}$ to construct $\mathcal{D}_{\text{SFT}}$ for supervised distillation, while the remaining data form $\mathcal{D}_{\text{RL}}$ for RL. Second, RL, detailed in §3.3.3, aligns the model's verdicts with correctness signals and yields interpretable step-level rewards for long-horizon decision-making. Together, these stages enable WebArbiter to provide robust, interpretable, and scalable supervision for web agents.

### 3.3.2 STAGE 1: REASONING DISTILLATION

Directly prompting an instruction-tuned LLM as a reward model often yields superficial, inconsistent chains that do not justify why an action advances the task. We therefore distill principle-guided reasoning from a stronger teacher. Concretely, o3 synthesizes structured justifications that first derive task-specific principles from the instruction and state, then ground these principles in the page, compare candidate actions against them, and finally output the preferred action. This equips WebArbiter with principles rather than surface heuristics. Ablations in §5.1.3 show that removing explicit principles and relying solely on reasoning-based justifications notably degrades performance, highlighting the role of principle induction in stabilizing step-level judgments. Given $(x^{(i)}, y^{(i)}) \in \mathcal{D}_{\text{SFT}}$, the teacher generates a justification $\hat{j}^{(i)} = (\hat{j}_1^{(i)}, \ldots, \hat{j}_{L_i}^{(i)})$. The distillation dataset is then: $\mathcal{D}_{\text{SFT}} = \{x^{(i)}, \hat{j}^{(i)}\}_{i=1}^K$.

**Objective.** Reasoning distillation adjusts $\theta$ to maximize the likelihood of generating the teacher justification $\hat{j}$ that concludes with the preferred action $y$ given $x$. We minimize the standard negative log-likelihood:

$$\mathcal{L}_{\text{SFT}}(\theta) = -\frac{1}{K} \sum_{i=1}^K \sum_{l=1}^{L_i} \log \pi_\theta \left( \hat{j}_l^{(i)} \mid x^{(i)}, \hat{j}_{<l}^{(i)} \right). \tag{5}$$

### 3.3.3 STAGE 2: REINFORCEMENT LEARNING

While distillation provides initial reasoning ability, it inherits teacher biases and may overfit to superficial patterns, limiting generalization to unseen environments. To further enhance judgment accuracy, stability, and generalization, we introduce a RL stage. WebArbiter $\pi_\theta$ is treated as a judgment policy that outputs a justification $j$ that concludes with a final verdict $\hat{y}$. During rollout,

Table 1: Data distribution of WEBPRMBENCH, the first comprehensive evaluation benchmark spanning diverse environments for WebPRMs.

| Environment | Mind2Web | | | WebArena | AssistantBench | WorkArena | Total |
|---|---|---|---|---|---|---|---|
| | Cross-Task | Cross-Website | Cross-Domain | | | | |
| **Count** | 142 | 148 | 417 | 201 | 30 | 212 | 1150 |

$\pi_\theta$ generates the full justification and verdict, after which a correctness reward $R(x, \hat{y}) \in \{-1, 1\}$ is assigned solely based on whether $\hat{y}$ matches the ground-truth preference $y$. The distilled model from §3.3.2 serves as the reference policy $\pi_{\text{ref}}$, ensuring stable optimization.

**Objective.** RL adjusts $\theta$ to maximize the expected reward while stabilizing reasoning style via KL regularization. The optimization objective is defined as:

$$\mathcal{L}_{\text{RL}}(\theta) \;=\; \max_{\pi_\theta} \; \mathbb{E}_{(x,y)\sim\mathcal{D}_{\text{RL}}, \, \hat{y}\sim\pi_\theta(j|x)} \Big[ R(x, \hat{y}) \Big] - \beta \, \mathbb{D}_{\text{KL}}(\pi_\theta \,\|\, \pi_{\text{ref}}). \tag{6}$$

In practice, we adopt Group Relative Policy Optimization (GRPO) (Shao et al., 2024) to optimize this objective, which enables stable updates under binary verifiable rewards. Through this RL stage, WebArbiter directly aligns its verdicts with correctness signals and converts structured justifications into reliable, interpretable step-level reward signals.

## 4 WEBPRMBENCH

This section introduces WEBPRMBENCH, a multi-environment benchmark for evaluating WebPRMs.

### 4.1 BENCHMARK CONSTRUCTION

WEBPRMBENCH is constructed from sucessful trajectories in AGENTREWARDBENCH (Lù et al., 2025), expanding beyond WEBREWARDBENCH (Chae et al., 2025), which only provides Mind2Web (Deng et al., 2023) and limited WebArena data (Zhou et al., 2023). We enrich WebArena with additional trajectories and incorporate AssistantBench (Yoran et al., 2024) and WorkArena (Drouin et al., 2024; Boisvert et al., 2025), resulting in broader coverage of real-world tasks across four environments. Mind2Web emphasizes cross-task generalization across heterogeneous websites. WebArena provides controlled environments such as shopping, CMS, Reddit, and GitLab. AssistantBench introduces open-world tasks on real websites. WorkArena focuses on enterprise workflows, including IT and HR. This diversity enables systematic evaluation across both consumer-facing and enterprise scenarios, covering a broad range of task complexities.

For each state, the action from the successful trajectory is retained as the positive label, and four rejected alternatives with associated reasoning traces are synthesized to form preference pairs. To ensure data quality, we sample negatives from diverse policy models to broaden coverage, apply rule-based filters to remove invalid or mismatched actions, discard inconsistent cases, and conduct expert verification to ensure reliability. We also conduct targeted auditing to eliminate false negatives. To avoid positional bias, the positive action is not fixed to a specific side and may appear on either side of the preference pair. Reasoning traces are truncated to a fixed length to minimize formatting noise. The resulting benchmark spans 1,150 step-level preference instances across four environments, as shown in Tab. 1. Full construction details and benchmark statistics are provided in Appendix E.

### 4.2 EVALUATION PROTOCOL

Evaluating WebPRMs requires metrics that capture both local preference fidelity and global decision reliability under realistic multi-candidate settings. Inspired by RMB (Zhou et al., 2025), we adopt two complementary metrics: *Pairwise Accuracy*, which measures correctness on preference pairs, and *Best-of-N (BoN) Accuracy*, which evaluates robustness when ranking among multiple distractors. Compared with *Pairwise Acc*, *BoN Acc* applies a stricter criterion by requiring the correct action to outrank all distractors simultaneously, providing stronger discriminative power and better alignment with downstream agent performance. A deeper analysis of *BoN* vs. *Pairwise Acc* is in Appendix F.

Table 2: Results on WEBPRMBENCH with *Pairwise* and *BoN Acc*. ★ denotes our models. Bold numbers indicate the best results, while underlined numbers denote the second best. Our WebArbiter-7B achieves the highest *Avg BoN Acc*, outperforming the second-best baseline, i.e., GPT-5, by 9.1.

| Models | Mind2Web | | WebArena | | AssistantBench | | WorkArena | | Avg. | |
|---|---|---|---|---|---|---|---|---|---|---|
| | *Pairwise* | *BoN* | *Pairwise* | *BoN* | *Pairwise* | *BoN* | *Pairwise* | *BoN* | *Pairwise* | *BoN* |
| *LLM-as-judge, Proprietary Language Models* | | | | | | | | | | |
| GPT-4o-mini | 81.74 | 50.92 | 78.23 | 56.72 | **89.17** | **73.33** | 81.43 | 46.70 | 82.64 | 56.92 |
| GPT-4o | 79.99 | 52.62 | 84.58 | 66.67 | 85.83 | 66.67 | **84.33** | 55.19 | 83.68 | 60.29 |
| GPT-5 | 80.86 | 62.39 | 84.83 | **71.64** | 81.67 | 63.33 | 81.14 | 64.62 | 82.13 | 65.50 |
| Claude-3.7-Sonnet | 80.20 | 57.90 | 82.80 | 64.10 | 81.50 | 61.30 | 82.10 | 60.60 | 81.65 | 60.98 |
| Gemini-2.5-Flash | 81.30 | 57.01 | 82.71 | 62.19 | 80.00 | 63.33 | 83.30 | 56.13 | 81.83 | 59.67 |
| DeepSeek-R1 | 81.62 | 57.37 | 82.04 | 60.21 | 78.49 | 56.18 | 84.12 | 63.89 | 81.57 | 59.41 |
| *LLM-as-judge, Open-source Language Models* | | | | | | | | | | |
| Qwen2.5-3B-Instruct | 76.46 | 36.93 | 60.32 | 15.42 | 75.83 | 33.33 | 64.45 | 19.34 | 69.27 | 26.76 |
| Qwen2.5-7B-Instruct | 77.79 | 39.18 | 74.88 | 42.79 | 84.17 | 53.33 | 77.58 | 35.85 | 77.61 | 42.78 |
| Llama-3-70B-Instruct | 80.55 | 49.36 | 77.36 | 50.75 | 85.83 | 70.00 | 79.08 | 40.09 | 80.71 | 52.55 |
| *WebPRMs (3B)* | | | | | | | | | | |
| WebShepherd-3B | 87.50 | 65.21 | 68.16 | 41.29 | 66.67 | 46.67 | 50.00 | 21.23 | 68.08 | 43.60 |
| ★ WebArbiter-3B | 93.32 | 78.42 | 81.97 | 56.22 | 78.33 | 46.67 | 81.01 | 54.81 | 83.65 | 59.06 |
| *WebPRMs (7B+)* | | | | | | | | | | |
| WebShepherd-8B | 86.66 | 73.69 | 68.33 | 43.88 | 55.92 | 30.00 | 54.56 | 25.53 | 64.34 | 43.28 |
| ★ WebArbiter-7B | **97.07** | **89.53** | **88.43** | 68.66 | **89.17** | 70.00 | 82.09 | **70.19** | **89.19** | **74.60** |

**Pairwise Acc.** Given a preference pair $(a^+, a^-)$, where $a^+$ is the correct action and $a^-$ is a rejected one, the WebPRM is correct if it assigns a higher preference to $a^+$. Formally:

$$Acc_{Pairwise} = \frac{1}{|\mathcal{D}_{\text{Bench}}|} \sum_{(a^+, a^-) \in \mathcal{D}_{\text{Bench}}} \mathbb{1}\left[\pi_\theta(a^+) \succ \pi_\theta(a^-)\right]. \tag{7}$$

**BoN Acc.** For each instance $(a^+, a^{-1}, \ldots, a^{-Q}) \in \mathcal{D}_{\text{Bench}}$, the WebPRM is considered correct only when $a^+$ is consistently ranked above all $Q$ distractors, with $Q = 4$ in our benchmark. BoN Acc is:

$$Acc_{BoN} = \frac{1}{|\mathcal{D}_{\text{Bench}}|} \sum_{i=1}^{|\mathcal{D}_{\text{Bench}}|} \prod_{q=1}^{Q} \mathbb{1}\left[\pi_\theta(a_i^+) \succ \pi_\theta(a_i^{-q})\right]. \tag{8}$$

## 5 EXPERIMENTS

We conduct comprehensive experiments to evaluate WebArbiter on the reward modeling benchmark WEBPRMBENCH in § 5.1 and on practical applications in § 5.2.

### 5.1 WEBPRMBENCH

#### 5.1.1 EXPERIMENTAL SETUP

**Baselines.** We compare WebArbiter against three categories of baselines. (1) Proprietary LLM-as-judge models, including GPT-4o-mini (OpenAI, 2024a), GPT-4o (OpenAI, 2024b), GPT-5 (OpenAI, 2025b), Claude-3.7-Sonnet (Anthropic, 2025), Gemini-2.5-Flash (Pichai & Hassabis, 2025), and DeepSeek-R1 (Guo et al., 2025a), which are prompted to act as judges by selecting the preferred action given task context. (2) Open-source LLM-as-judge models, represented by Qwen2.5-3B-Instruct and Qwen2.5-7B-Instruct (Qwen et al., 2025), and Llama-3-70B-Instruct (Meta, 2024), providing accessible yet competitive instruction-tuned baselines. (3) WebPRMs, where we include WebShepherd (Chae et al., 2025).

**Implementation Details.** We train WebArbiter on WEBPRM Collection (Chae et al., 2025), which comprises 30k step-level preference pairs drawn from the Mind2Web environment. We use 10k pairs for stage-1 reasoning distillation and the remainder for stage-2 RL. Models are initialized from

Table 3: Ablation results on WEBPRMBENCH with Qwen2.5-7B-Instruct as backbone. We report *Pairwise* and *BoN Acc* across web environments. WebArbiter, combining principle-guided reasoning distillation with RL, achieves the highest overall performance.

| Method | Mind2Web | | WebArena | | AssistantBench | | WorkArena | | Avg. | |
|---|---|---|---|---|---|---|---|---|---|---|
| | *Pairwise* | *BoN* | *Pairwise* | *BoN* | *Pairwise* | *BoN* | *Pairwise* | *BoN* | *Pairwise* | *BoN* |
| Instruct (Original) | 77.79 | 39.18 | 74.88 | 42.79 | 84.17 | 53.33 | 77.58 | 35.85 | 77.61 | 42.78 |
| Instruct + Cold Start RL | 96.18 | 86.00 | 71.10 | 35.80 | 72.40 | 33.60 | 74.90 | 37.90 | 78.15 | 48.33 |
| Instruct + Cold Start RL + Principles | 96.18 | 88.00 | 77.80 | 46.30 | 80.10 | 48.90 | 82.40 | 51.80 | 84.12 | 58.75 |
| Instruct + SFT$_{w/o\ Principles}$ + RL | 98.48 | 94.34 | 74.60 | 41.50 | 77.20 | 40.20 | 79.10 | 44.60 | 82.35 | 55.16 |
| ★ WebArbiter-7B | 97.07 | 89.53 | 88.43 | 68.66 | 89.17 | 70.00 | 82.09 | 70.19 | 89.19 | 74.60 |

Qwen2.5-3B-Instruct and Qwen2.5-7B-Instruct (Qwen et al., 2025) and fine-tuned with LoRA (Hu et al., 2022). Further implementation details are provided in the Appendix C, and all prompts are provided in Appendix D.

**Evaluation Metrics.** We report results using two complementary metrics: *Pairwise Accuracy*, which measures correctness on individual preference pairs, and *Best-of-N (BoN) Accuracy*, which evaluates robustness under multi-candidate settings. Detailed definitions are provided in § 4.2.

### 5.1.2 MAIN RESULTS

**WebArbiter Significantly Outperforms Baselines.** As shown in Tab. 2, WebArbiter achieves the highest *Avg. Pairwise Acc* and *Avg. BoN Acc*, surpassing both proprietary and open-source LLMs. While LLM-as-judge methods often maintain moderate *Pairwise Acc*, their performance drops sharply on *BoN Acc*, revealing poor robustness to hard negatives. In contrast, WebArbiter sustains strong results on both metrics, establishing its reliability under realistic multi-candidate settings. We show that the proposed training pipeline generalizes across backbone families in Appendix G, where WebArbiter-8B$_{Qwen3}$ achieves the highest Avg. BoN Acc of 76.66%, and provide additional analyses of inference-time scaling in Appendix H.

**Advantage over the SOTA WebPRM.** WebShepherd (Chae et al., 2025) represents the previous SOTA WebPRMs. Trained on the same WEBPRM Collection, which was drawn from the Mind2Web environment, WebArbiter-7B achieves an *Avg. BoN Acc* of 74.60%, surpassing WebShepherd-8B by an absolute gain of 31%. Unlike WebShepherd, which relies on fragile checklists, WebArbiter employs principle-guided reasoning, yielding judgments robust to environment and page variations. Case studies illustrating these differences are provided in Appendix I.

**Robust Generalization Across Environments.** As shown in Tab. 2, it attains SOTA *BoN Acc* on Mind2Web and WorkArena, and remains competitive with strong proprietary LLMs on WebArena and AssistantBench. These results indicate that principle-guided reasoning enables both effective in-domain learning and robust performance in heterogeneous, noisy, and enterprise-scale environments.

### 5.1.3 ANALYSIS OF TRAINING DESIGN

**Training Recipes** We compare four training variants to disentangle the effects of RL, principle guidance, and justification style. *Instruct (Original)* denotes a purely instruction-tuned model without additional optimization. *Instruct + Cold Start RL* directly applies RL on top of the instruction model. *Instruct + Cold Start RL + Principles* augments RL with principle prompting during training, enabling explicit principle induction before judgment. *Instruct + SFT$_{w/o\ Principles}$ + RL* performs reasoning distillation without principles, followed by RL, thereby testing whether narrative-style justifications alone are sufficient. As shown in Tab. 3, WebArbiter achieves the best performance. Explicit principles anchor judgments to progress, producing stable supervision under multi-candidate web settings.

**RL Alone is Unstable Across Web Environments.** *Cold Start RL* performs well on in-domain Mind2Web but collapses on out-of-domain benchmarks. This highlights that reward optimization without reasoning distillation struggles in noisy and complex environments.

**Principles Enable Cross-Environment Generalization.** Augmenting RL with principles improves both *Avg. Pairwise* and *BoN Acc*, especially on AssistantBench and WorkArena, where real-world

Table 4: Results on WEBPRMBENCH under full-data and limited-data (10K) training regimes. We report *Pairwise* and *BoN Acc* across web environments. Reasoning distillation improves over answer-only SFT, while WebArbiter, i.e., reasoning distillation + RL, achieves the best overall performance.

| Method | Mind2Web | | WebArena | | AssistantBench | | WorkArena | | Avg. | |
|---|---|---|---|---|---|---|---|---|---|---|
| | *Pairwise* | *BoN* | *Pairwise* | *BoN* | *Pairwise* | *BoN* | *Pairwise* | *BoN* | *Pairwise* | *BoN* |
| *Train on Full Data* | | | | | | | | | | |
| Instruct + SFT | 85.14 | 60.91 | 80.85 | 52.73 | 82.50 | 56.67 | 79.57 | 52.88 | 82.02 | 55.80 |
| Instruct + Distilled + SFT | 87.42 | 61.18 | 81.59 | 52.73 | 83.33 | 63.33 | 81.13 | 56.73 | 83.37 | 58.49 |
| ★ WebArbiter-7B (Instruct + Distilled + RL) | 97.07 | 89.53 | 88.43 | 68.66 | 89.17 | 70.00 | 82.09 | 70.19 | 89.19 | 74.60 |
| *Train on 10K (Stage-1 Reasoning Distillation) Data* | | | | | | | | | | |
| Instruct + SFT | 84.53 | 60.82 | 82.21 | 58.71 | 82.50 | 56.67 | 80.58 | 39.62 | 82.46 | 53.96 |
| Instruct + Distilled | 85.20 | 63.40 | 83.10 | 61.80 | 83.00 | 60.20 | 81.40 | 55.60 | 83.18 | 60.25 |

tasks require context- and state-dependent judgments beyond surface layout cues. AssistantBench features open-world websites with high structural variability, while WorkArena involves enterprise workflows governed by state-dependent constraints. Principle-guided reasoning provides transferable criteria for assessing true task progress in both cases, improving robustness and generalization.

**Reasoning Without Principles is Insufficient.** $SFT_{w/o\ Principles} + RL$, which relies solely on narrative-style justifications, improves linguistic fluency and coherence of the generated explanations but consistently underperforms principle-aware settings. Without explicit principles to anchor judgment, the model tends to rationalize actions post hoc based on surface plausibility, making it vulnerable to spurious correlations and context-specific cues. As a result, narrative reasoning alone is insufficient to reliably track genuine task progress in complex, long-horizon real-world web navigation.

**Reasoning Supervision** We analyze the role of reasoning supervision by comparing answer-only SFT, distilled reasoning, and RL under both full-data and limited-data settings. *Instruct + SFT* optimizes the instruction-tuned model to directly output the final preference decision, without exposing any intermediate reasoning or justification during training. *Instruct + Distilled + SFT* runs an answer-only SFT stage on top of the distilled checkpoint, fine-tuning the model directly toward the final decision and serving as a controlled comparison to RL-based training. *WebArbiter (Instruct + Distilled + RL)* further builds upon distilled reasoning by applying RL with verifiable rewards, encouraging principle-guided judgments that better reflect true task progress. Results on WEBPRMBENCH are reported in Tab. 4.

**Reasoning Distillation Improves Judgment Stability, with RL as an Amplifier.** Comparing *Instruct + Distilled + SFT* with *Instruct + SFT*, we find that reasoning supervision leads to more reliable reward judgments, particularly in multi-candidate settings measured by *BoN Acc*. Under the full-data setting, applying answer-only SFT after distillation yields environment-dependent gains, as final-answer optimization can reintroduce shortcut correlations specific to individual web environments. Nevertheless, reasoning distillation induces a more stable discrimination among competing trajectories by grounding judgments in true task progress rather than surface-level cues. Building upon this reasoning distillation phase, WebArbiter further applies RL to enlarge the margin between truly progress-making and spurious trajectories, achieving the highest overall performance.

**Reasoning Supervision Is Especially Effective Under Limited Data.** Under the 10K (Stage-1 Reasoning Distillation) setting, *Instruct + Distilled* consistently outperforms *Instruct + SFT* across all environments, yielding clear improvements in both *Pairwise* and *BoN Acc*. Since both models are trained with identical data budgets, these gains cannot be attributed to data scale, but instead reflect a training objective that explicitly biases the model toward progress-aware reward judgments.

## 5.2 REWARD-GUIDED TRAJECTORY SEARCH

### 5.2.1 EXPERIMENTAL SETUP AND IMPLEMENTATIONS

Reward-guided trajectory search represents one of the most practical applications of PRMs, as it directly leverages fine-grained step-level supervision to improve decision quality during agent execution. To evaluate WebArbiter in this setting, we conduct experiments on WebArena-Lite (Liu et al., 2024b), which contains diverse, long-horizon tasks such as online shopping and content

Table 5: Success Rates (%) of trajectory search with GPT-4o-mini and GPT-4o as policy on WebArena-Lite. [*] Results reported from the WebShepherd (Chae et al., 2025). $\Delta$ is relative to the *w/o Trajectory Search* baseline. Our WebArbiter consistently achieves the highest gains across both policy models.

| Policy | WebPRM | Shopping | CMS | Reddit | GitLab | MAP | Avg. | $\Delta$ |
|--------|--------|----------|-----|--------|--------|-----|------|----------|
| GPT-4o-mini | w/o Trajectory Search[*] | 21.74 | 22.86 | 19.05 | 34.38 | 19.35 | 23.48 | – |
| | GPT-4o-mini | 24.44 | 22.86 | 26.32 | 33.33 | 15.38 | 24.47 | +0.99 |
| | WebShepherd-8B[*] | 26.09 | **45.71** | 23.81 | 40.62 | 35.48 | 34.34 | +10.86 |
| | ★ WebArbiter-7B | **37.78** | 42.86 | **36.84** | **46.67** | **38.46** | **40.52** | **+17.04** |
| GPT-4o | w/o Trajectory Search[*] | 23.91 | 31.43 | 28.57 | 56.25 | 19.35 | 31.90 | – |
| | GPT-4o-mini | 26.67 | 37.14 | 42.11 | 40.00 | 19.23 | 33.03 | +1.13 |
| | WebShepherd-8B[*] | 30.43 | **42.86** | 47.62 | 46.88 | 35.48 | 40.65 | +8.75 |
| | ★ WebArbiter-7B | **44.44** | **42.86** | **52.63** | **56.67** | **38.46** | **47.01** | **+15.11** |

management, closely reflecting real-world web activities. Performance is measured with Success Rate. Following WebShepherd (Chae et al., 2025), we adopt a Best-of-N sampling strategy: the policy model generates $N = 5$ candidate actions for each step, and WebArbiter selects the most promising one through a Knockout Tournament mechanism (Guo et al., 2025b). We evaluate two policies, GPT-4o-mini (OpenAI, 2024a) and GPT-4o (OpenAI, 2024b).

### 5.2.2 DOWNSTREAM ANALYSIS ACROSS DOMAINS

As shown in Tab. 5, WebArbiter achieves substantial average improvements under both policy models, significantly outperforming all baselines. These gains stem from two main factors. First, reasoning mitigates spurious correlations that often mislead WebPRMs in domains such as Shopping and Reddit. The improvements on Shopping are particularly pronounced, as these tasks require dense semantic retrieval and inference: stronger policies can propose more promising candidate actions, and WebArbiter's structured reward modeling further amplifies these advantages. Second, in GitLab, tasks frequently admit multiple equivalent paths. WebShepherd is brittle under such variability, whereas WebArbiter reasons over historical trajectories and the current state to evaluate action validity, enabling stronger generalization in dynamic workflows. We provide qualitative case studies in Appendix I to further illustrate these failure modes of checklist-based supervision. MAP presents a distinct pattern: tasks primarily involve information retrieval over OpenStreetMap, where success hinges on the semantic precision of search queries. Naïve reward signals fail to assess whether a query targets the intended geographic entity, while WebArbiter nearly doubles the baseline by reasoning over query-task alignment. By contrast, CMS exhibits a more template-driven structure, where actions closely follow standardized patterns. In such settings, checklist-based supervision remains comparatively effective, which narrows the relative performance gap. Overall, WebArbiter's reasoning-first design consistently provides robust, interpretable, and scalable supervision across diverse domains.

## 6 CONCLUSION

We presented WebArbiter, a reasoning-first, principle-inducing process reward model that frames reward modeling as structured text generation and produces auditable step-level judgments with rationales. Through reasoning distillation and RL, WebArbiter converts superficial correlations into robust, progress-aware signals that verify task advancement, yield consistent step-level judgments across trajectories, and generalize across dynamic web environments. To support systematic evaluation, we released WEBPRMBENCH, the first comprehensive evaluation benchmark spanning diverse environments for WebPRMs in web navigation, covering four domains with diverse tasks and step-level preference annotations. Experiments demonstrate SOTA performance on WEBPRMBENCH and improvements in reward-guided trajectory search on WebArena-Lite, establishing principle-guided reasoning WebPRMs as a robust and interpretable foundation for scalable web agents.

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

# Contents

# A NOTATION SUMMARY

For clarity, we summarize the main notations used throughout this paper:

- $\mathcal{E}$: web environment, defined by state space $\mathcal{S}$, action space $\mathcal{A}$, and observation space $\mathcal{O}$.
- $T$: state transition function $T : \mathcal{S} \times \mathcal{A} \to \mathcal{S}$.
- $\mathcal{I}$: task instruction.
- $s_p, o_p, a_p$: state, observation, and action at step $p$.
- $c_p$: reasoning trace associated with action $a_p$.
- $c_{<p}$: reasoning traces of all previously executed actions.
- $\tau = (a_1, \ldots, a_P)$: trajectory of length $P$.
- $j = (j_1, \ldots, j_L)$: structured justification of length $L$, consisting of explicit reasoning and a final verdict.
- $\pi_\theta$: WebArbiter model parameterized by $\theta$.
- $\hat{y}$: predicted preference verdict.
- $\mathcal{D}_{\text{Train}}, \mathcal{D}_{\text{SFT}}, \mathcal{D}_{\text{RL}}$: training datasets for supervised distillation and RL.
- $\mathcal{D}_{\text{Bench}}$: evaluation dataset for WEBPRMBENCH.
- $a^+$: action from a successful trajectory (positive label).
- $a^-$ or $a^{-q}$: rejected action ($q = 1, \ldots, Q$).
- $Q$: number of distractors in WEBPRMBENCH (set to $4$).
- $M$: dataset size (training or evaluation).
- $R(x, \hat{y})$: verifiable reward signal.

# B EXAMPLE OF PREFERENCE DATASET

To illustrate the construction of a preference pair, consider a task instruction:

- **Instruction** $\mathcal{I}$: "Find the 2026 conference submission page on the ICLR website.''
- **Current observation** $o_p$: [356] banner 'header', role='banner' [359] link 'Home' [380] button 'Select Year (2026)' [386] button 'Dates' [391] button 'Calls' [396] button 'Guides' [401] button 'Organization' [403] heading 'ICLR 2026'
- **History** $(a_{<p}, c_{<p})$:
  - $a_1$: Search "ICLR", $c_1$: "I need to find the official ICLR website first. Let me search for 'ICLR' to locate it.''
  - $a_2$: Click link "ICLR homepage", $c_2$: "I can see the ICLR official website link in the search results. Clicking on it will take me to the ICLR homepage where I can find the conference submission information.''
- **Candidate actions and reasoning traces**:
  - $(a_p^1, c_p^1)$: Click link "Call for Papers" ; $c_p^1$: "I can see a 'Call for Papers' link on the ICLR homepage. This link would likely lead to the submission details page, which should contain information about the 2026 conference submission process that I'm looking for.''
  - $(a_p^2, c_p^2)$: Click "About" link; $c_p^2$: "I can see an 'About' link on the ICLR homepage. Since I need to find the 2026 conference submission page, the 'About' section might contain conference overview information including links to submission details or important dates for the 2026 conference.''
- **Label** $y$: $a_p^1$ is preferred.

This example is represented in the dataset as:

$$(\mathcal{I}, o_p, a_{<p}, c_{<p}, (a_p^1, c_p^1), (a_p^2, c_p^2), y = a_p^1).$$

## C  TRAINING DETAILS

All training is conducted on 8 NVIDIA A100-80GB GPUs with fixed random seeds. Our training framework is based on LLama-Factory (Zheng et al., 2024) and VERL (Sheng et al., 2024). In addition to the Qwen2.5 (Qwen et al., 2025) backbones used in the main experiments, we also train WebArbiter on Qwen3 (Yang et al., 2025) backbones (4B and 8B); results are reported in Appendix G.

**Distillation Stage.**  For Qwen2.5 backbones (3B and 7B), we use a learning rate of 8e-4 with LoRA rank 128. For Qwen3-4B, the learning rate is 1e-3 with LoRA rank 96. For Qwen3-8B, the learning rate is 1e-3 with LoRA rank 64 and a reduced batch size of 64. All variants use a cosine learning rate scheduler with a warmup ratio of 0.1 and a maximum sequence length of 8,192 tokens. Models are trained for 5 epochs.

**RLVR Stage.**  We employ the VERL framework for GRPO training. For Qwen2.5 backbones, the learning rate is set to 7e-6 (7B) and 9e-6 (3B). For Qwen3 backbones (4B and 8B), the learning rate is 1e-5. The training uses a fixed batch size of 512 with a mini-batch size of 128, and adopts Fully Sharded Data Parallel (FSDP) for enhanced memory efficiency. For rollout generation, we deploy vLLM with tensor parallelism of 4 and GPU memory utilization limited to 0.4. Response sampling uses standard parameters (temperature=1.0, top-p=1.0). The rollout group size is $G = 7$ for Qwen2.5 backbones and $G = 12$ for Qwen3 backbones. We apply KL regularization with a coefficient of $1.0 \times 10^{-3}$ and a clip ratio of 0.2. The maximum input sequence length is 8,192 tokens, and the maximum response length is 4,096 tokens.

## D  PROMPT REPOSITORY

```
WebArbiter

You are a skilled expert at evaluating assistant responses. You
    should evaluate given responses based on the given judging
    criteria.\n Given the context of the conversation and two
    responses from the Assistant, you need to determine the better
    response. Provide an overall comprehensive comparison upon them.
#### Intent ####
{intent}
#### AXTREE ####
Note: [bid] is the unique alpha-numeric identifier at the
    beginning of lines for each element in the AXTree. Always use
    bid to refer to elements in your actions.
{observation}
#### Trajectory ####
Note: The trajectory contains the sequence of previous actions and
    their corresponding thoughts. Each entry reflects the agent's
    internal reasoning ('thought') and the concrete operation it
    performed ('action').
{trajectory}
#### start url ####
{start_url}
#### current url ####
The URL provides clues about the user's position in the
    application flow. Use both the path and query parameters to
    infer page type (e.g., homepage, search results, product
    detail, cart, checkout).
{current_url}
#### Assistant Responses ####
[The Begin of Response 1]
THOUGHT:
{thought1}
ACTION:
{action1}
```

```
[The End of Response 1]
[The Begin of Response 2]
THOUGHT:
{thought2}
ACTION:
{action2}
[The End of Response 2]
### Output Instructions ###
Format your output strictly using the following XML-style tags:
<State>Summarize the current state based on the URL, AXTree, and
    previous actions. Include what page the user is currently on,
    and what relevant UI elements or information are
    visible.</State>
<Criteria>Other potential criteria specific to the query and the
    context, and the weights of each criteria.</Criteria>
<Analysis>Compare Response 1 and Response 2 in detail according to
    the <State> and <Criteria>.</Analysis>
<Answer>Response 1 or Response 2</Answer>
Rules for <Answer>:
- If Response 1 is better, output exactly: <Answer>Response
    1</Answer>
- If Response 2 is better, output exactly: <Answer>Response
    2</Answer>
Important Notes:
- Be objective and base your evaluation strictly on the content of
    the responses.
- Do not let the response order, length bias your judgment.
```

## E  BENCHMARK CONSTRUCTION

### E.1  PREFERENCE PAIR CONSTRUCTION

**Positive samples.**  We construct WEBPRMBENCH using the successful trajectories from AGEN-TREWARDBENCH, a human-verified evaluation suite that aggregates over a thousand trajectories generated by multiple LLM-based web agents across diverse real-world environments. Each trajectory in AGENTREWARDBENCH is annotated for success and execution quality by expert annotators, providing a reliable source of environment-grounded optimal behavior. From this dataset, we select only those trajectories that complete each task with the minimum number of steps. Each trajectory is independently reviewed by annotators to ensure monotonic progress and to verify that no redundant or detour actions are present. When deviations are identified, annotators revise the trajectory to recover the shortest valid execution path consistent with successful task completion. For consistency, missing reasoning traces are completed to ensure that every state–action pair is paired with a coherent rationale. The resulting actions from these validated minimal-step trajectories serve as positive labels, reflecting actions empirically verified to succeed in the real web environment.

**Negative samples.**  For each state, we sample four alternative actions and their associated reasoning from a diverse ensemble of policy models, covering both open-source and proprietary LLMs. The pool includes high-capacity instruction-tuned models such as Qwen2.5-7B / 72B-Instruct (Qwen et al., 2025), Llama-3.3-8B / 70B-Instruct (Meta, 2024), as well as frontier commercial models including GPT-4o / 4o-mini (OpenAI, 2024a;b), Claude-3.5-Haiku / Claude-3.7-Sonnet (Anthropic, 2024b; 2025), and Gemini-2.5-Flash / Gemini-2.5-Pro (Comanici et al., 2025). This ensures that alternative actions exhibit broad stylistic and policy diversity rather than reflecting any single model's reasoning behavior. Since alternative actions may still succeed under certain web interfaces, we apply a rule-based filtering procedure to remove actions that remain potentially valid. We retain only actions that are clearly invalid or non-progressing, ensuring that negative samples correspond to failures under the actual environment dynamics rather than differences in reasoning style. To ensure consistency and avoid false negatives, the filtered actions are manually reviewed, and any remaining actions that appear potentially valid are discarded. If more than four valid rejected actions

Table 6: WEBPRMBENCH Website Visit Counts

| Domain | # | Domain | # | Domain | # |
|---|---|---|---|---|---|
| service-now.com | 212 | wa-openstreetmap-xl-1 | 48 | wa-forum-xl-2 | 38 |
| wa-gitlab-xl-1 | 23 | wa-shopping-admin-xl-1 | 21 | google.com | 17 |
| wa-openstreetmap-xl-2 | 17 | wa-shopping-admin-xl-2 | 16 | ryanair.com | 12 |
| wa-forum-xl-1 | 12 | wa-shopping-xl-2 | 11 | last.fm | 10 |
| delta.com | 9 | duckduckgo.com | 8 | wa-gitlab-xl-2 | 8 |
| redbox.monster | 7 | target.com | 7 | united.com | 7 |
| wa-shopping-xl-1 | 7 | kohls.com | 6 | soundcloud.com | 6 |
| spothero.com | 6 | yellowpages.com | 6 | amctheatres.com | 5 |
| exploretock.com | 5 | qatarairways.com | 5 | aa.com | 4 |
| foxsports.com | 4 | ikea.com | 4 | kayak.com | 4 |
| marriott.com | 4 | rentalcars.com | 4 | sixflags.com | 4 |
| travelzoo.com | 4 | yelp.com | 4 | discogs.com | 3 |
| gamestop.com | 3 | koa.com | 3 | mta.info | 3 |
| tesla.com | 3 | cabelas.com | 2 | rottentomatoes.com | 2 |
| extremeweatherwatch.com | 1 | | | | |

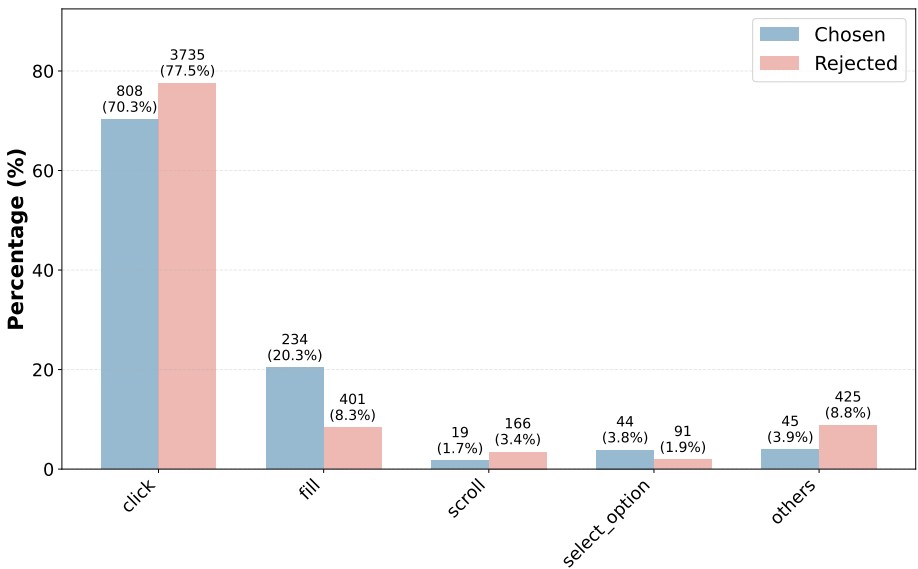

Figure 3: Action-type distribution in WEBPRMBENCH.

remain after filtering, we randomly sample a subset to maintain a consistent number of action pairs per instance. All rationales are truncated to a fixed length to reduce formatting noise while preserving semantic content.

### E.2 DATASET COMPOSITION AND STATISTICS

The final benchmark consists of 1,150 step-level preference instances across four environments, each containing one environment-verified positive action and four negative alternatives.

**Website distribution.** Tab. 6 summarizes the distribution of visited websites in WEBPRMBENCH, highlighting the diversity and long-tailed nature of real-world web environments covered by the benchmark.

**Action-type distribution.** Fig. 3 reports the action-type distributions of environment-verified positive actions and negative actions in WEBPRMBENCH. In both sets, click and fill constitute the majority of actions, consistent with common interaction primitives in real-world web navigation. The distribution of rejected actions closely mirrors that of chosen actions, with only minor shifts in

Table 7: Standard deviation of model scores under *BoN* and *Pairwise* evaluation across web environments on WEBPRMBENCH.

|  | Std. deviation |
| --- | --- |
| Mind2Web-BoN | 0.149 |
| Mind2Web-pairwise | 0.060 |
| WebArena-BoN | 0.153 |
| WebArena-pairwise | 0.081 |
| AssistantBench-BoN | 0.139 |
| AssistantBench-pairwise | 0.093 |
| WorkArena-BoN | 0.173 |
| WorkArena-pairwise | 0.116 |

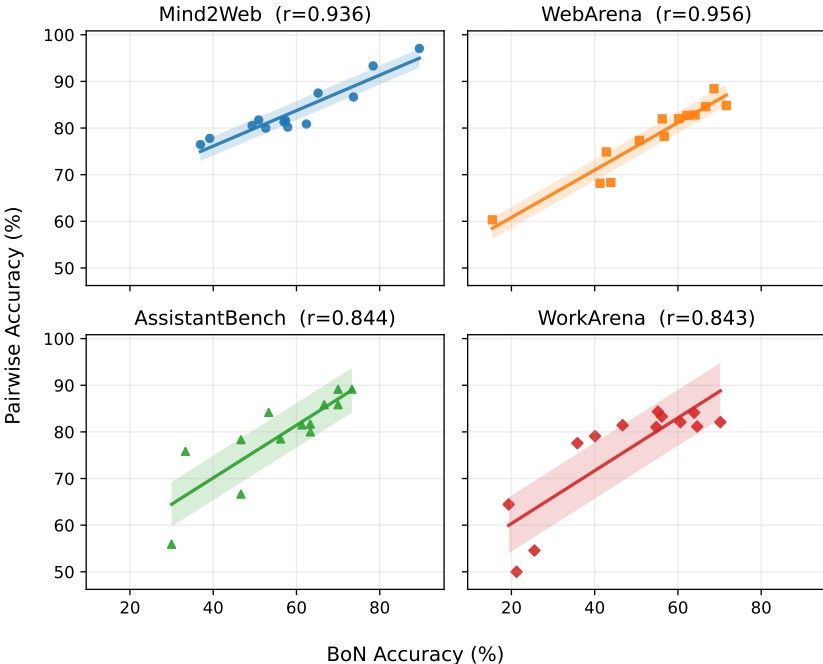

Figure 4: Correlation between *BoN* and *Pairwise Acc* across web benchmarks. Each scatter point corresponds to a PRM. We report the correlation coefficient $r$ for each environment. While the two metrics are strongly correlated across all environments, BoN exhibits higher variance and provides finer-grained discrimination among models, particularly in complex web environments.

relative proportions, indicating that negative actions are not dominated by rare or structurally distinct types but arise from the same high-frequency operations as successful actions. As a result, action-type identity alone provides no reliable signal for correctness. Effective discrimination, therefore, requires assessing whether an action advances task progress under the current state, rather than relying on action-type assumptions or global frequency-based heuristics.

## F  ANALYSIS OF *BoN Acc* VS. *Pairwise Acc* EVALUATION

We analyze how *BoN Acc* and *Pairwise Acc* behave as evaluation metrics for WebPRMs on WEBPRM-BENCH. This comparison is practically important because WebPRMs are commonly used to rank multiple candidate actions during agent execution, whereas *Pairwise Acc* only measures correctness on isolated preference pairs. In our benchmark, *BoN Acc* imposes a stricter evaluation criterion by requiring the correct action to outperform all distractors simultaneously, making it more representative of realistic multi-candidate decision-making scenarios.

Table 8: Results on WEBPRMBENCH with additional Qwen3 backbones. Models marked with ★ are ours.

| Models | Mind2Web | | WebArena | | AssistantBench | | WorkArena | | Avg. | |
|---|---|---|---|---|---|---|---|---|---|---|
| | *Pairwise* | *BoN* | *Pairwise* | *BoN* | *Pairwise* | *BoN* | *Pairwise* | *BoN* | *Pairwise* | *BoN* |
| *3~4B* | | | | | | | | | | |
| WebShepherd-3B | 87.50 | 65.21 | 68.16 | 41.29 | 66.67 | 46.67 | 50.00 | 21.23 | 68.08 | 43.60 |
| ★ WebArbiter-3B$_{Qwen2.5}$ | 93.32 | 78.42 | 81.97 | 56.22 | 78.33 | 46.67 | 81.01 | 54.81 | 83.65 | 59.06 |
| ★ WebArbiter-4B$_{Qwen3}$ | **98.55** | **94.73** | **83.21** | **61.19** | **92.50** | **83.33** | 76.68 | 50.96 | **87.73** | **72.55** |
| *7~8B* | | | | | | | | | | |
| WebShepherd-8B | 86.66 | 73.69 | 68.33 | 43.88 | 55.92 | 30.00 | 54.56 | 25.53 | 64.34 | 43.28 |
| ★ WebArbiter-7B$_{Qwen2.5}$ | 97.07 | 89.53 | **88.43** | **68.66** | 89.17 | 70.00 | 82.09 | **70.19** | 89.19 | 74.60 |
| ★ WebArbiter-8B$_{Qwen3}$ | 98.33 | 94.09 | 86.92 | 67.16 | **92.50** | **80.00** | **86.66** | 65.38 | **91.10** | **76.66** |

***BoN Acc* Provides Stronger Discriminative Power Across Environments.** Tab. 7 reports the standard deviation of model scores under *BoN* and *Pairwise Acc*. Across all four environments, *BoN Acc* consistently exhibits higher variance than *Pairwise Acc*, indicating substantially less score compression and larger separation among models. This effect is particularly pronounced in WorkArena, where complex interaction dynamics and harder distractors amplify small weaknesses into measurable performance gaps. These results confirm that *BoN Acc* offers finer-grained discrimination among WebPRMs, especially in settings where robust multi-candidate judgment is required.

***BoN Acc* and *Pairwise Acc* Are Consistent but Not Equivalent.** Fig. 4 shows that *BoN Acc* and *Pairwise Acc* are strongly positively correlated across all environments. This indicates that the two metrics capture broadly aligned notions of WebPRM quality and induce similar overall ordering of models. However, the correlation strength varies across environments, reflecting differences in interaction structure and distractor difficulty.

## G   GENERALIZATION ACROSS BACKBONE FAMILIES

To verify that the proposed two-stage training pipeline is not tied to a specific model family, we train WebArbiter on Qwen3 (Yang et al., 2025) backbones (4B and 8B) in addition to the Qwen2.5 backbones reported in the main paper. All variants use the same training data, distillation strategy, and RL procedure; only backbone-specific hyperparameters differ (see Appendix C).

As shown in Tab. 8, WebArbiter achieves strong performance on both backbone families. Within the 3~4B group, WebArbiter-4B$_{Qwen3}$ substantially outperforms WebArbiter-3B$_{Qwen2.5}$ across all environments, improving *Avg. BoN Acc* from 59.06% to 72.55%. This result approaches WebArbiter-7B$_{Qwen2.5}$ (74.60%) with roughly half the parameters, suggesting that stronger base models amplify the benefits of principle-guided reasoning distillation. Within the 7~8B group, WebArbiter-8B$_{Qwen3}$ achieves the highest *Avg. BoN Acc* of 76.66%, outperforming WebArbiter-7B$_{Qwen2.5}$ by 2.06 points.

Overall, these results confirm that the proposed two-stage training pipeline generalizes across model families and benefits from stronger base models without requiring pipeline modifications.

## H   INFERENCE-TIME SCALING

We further analyze how WebArbiter benefits from increased inference-time compute by varying the number of sampled reward evaluations. As shown in Fig. 5, both *Pairwise* and *BoN Acc* improve consistently as the sampling budget increases for WebArbiter-3B and WebArbiter-7B, confirming that the proposed reasoning-based WebPRM supports inference-time scaling. The improvements are moderate for *Pairwise Acc* but substantially more pronounced under the stricter *BoN Acc*, highlighting the advantage of additional inference computation in multi-distractor ranking scenarios.

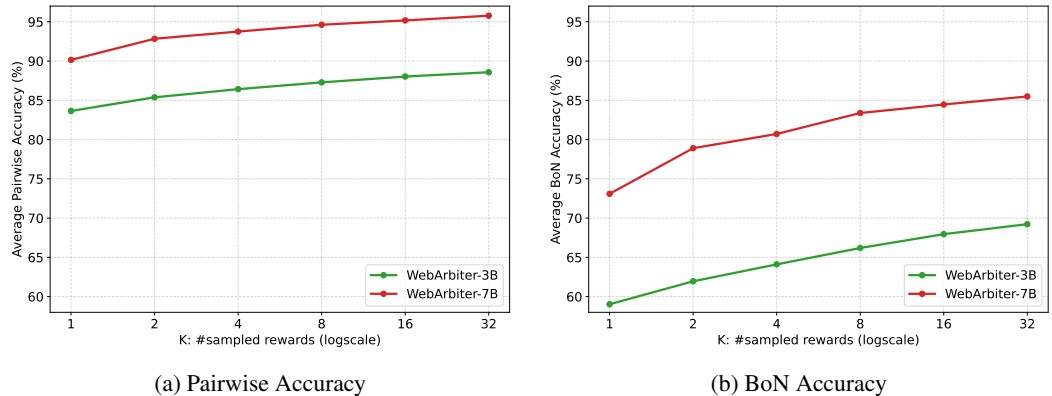

(a) Pairwise Accuracy          (b) BoN Accuracy

Figure 5: Inference-time scaling of WebArbiter. **Left:** *Pairwise* and **Right:** *BoN Acc* as the number of sampled reward evaluations $K$ increases.

## I  CASE STUDY: WEBARBITER VS. WEBSHEPHERD

This section presents two GitLab-based case studies that concretely illustrate the failure modes of checklist-driven WebPRMs. These cases highlight how checklist-style supervision can become brittle under structural variability, and how WebArbiter's reasoning-based evaluation yields more reliable action preferences.

### I.1  MILESTONE CREATION UNDER MULTIPLE EQUIVALENT PATHS

The task is to create a milestone for an upcoming merge operation. At the current step, the agent is on the GitLab project homepage, where the left navigation menu exposes an "Issues" entry that directly supports milestone management, alongside other entries such as "Project information" that lead to alternative but non-essential paths. Two candidate actions are considered: navigating through "Project information" or directly entering "Issues", as shown in Fig. 6.

WebShepherd evaluates these candidates using checklist-style criteria that emphasize procedurally typical navigation patterns. In GitLab, however, multiple interface paths may lead to the same functionality, and conventionally expected steps are not always necessary in the current context. As a result, WebShepherd may favor navigating through "Project information" despite the fact that milestone creation is already accessible via "Issues", introducing an avoidable detour. In contrast, WebArbiter reasons over the current state and task objective to assess whether an action directly contributes to task progress. Observing that the required functionality is already available, it assigns higher preference to entering "Issues" and deprioritizes redundant navigation steps. This example reflects a common characteristic of GitLab workflows: path multiplicity with varying informational value, under which checklist-driven supervision struggles to generalize consistently.

### I.2  MERGE REQUEST IDENTIFICATION UNDER AMBIGUOUS CONTEXT

The second task requires locating a specific merge request referenced by a 404 link, checking for a reply, and responding accordingly. The agent is initially presented with a merge request overview page listing multiple candidates, none of which are explicitly linked to the given URL, while a global search function is available to resolve this ambiguity, as shown in Fig. 7. The agent can either open one of the visible merge requests or initiate a search to identify the correct target.

Checklist-based supervision tends to favor actions that satisfy immediate procedural milestones, such as entering a merge request page, without explicitly verifying whether the selected entity matches the task specification. Consequently, opening an arbitrary merge request may be preferred even though the task's referent has not yet been identified. WebArbiter, by contrast, evaluates action validity by reasoning about task preconditions and required evidence. Since identifying the correct merge request is a prerequisite for any subsequent review or response, actions that do not support disambiguation are penalized. WebArbiter therefore prefers initiating a search and defers content-level interaction

until the task context is correctly grounded. This case further illustrates how checklist-based rewards can conflate interaction progress with task progress in dynamic settings, whereas reasoning-based evaluation maintains alignment between actions and task intent.

## J  FAILURE CASE ANALYSIS

While Appendix I illustrates the advantages of reasoning-based evaluation over checklist-driven supervision, we complement this analysis by examining two recurring failure patterns observed during reward-guided trajectory search on GitLab. Both cases reveal open challenges shared by text-based WebPRMs that rely on accessibility-tree observations.

### J.1  SAFE-ACTION BIAS

The task is to check personal todos. At the current step, the agent is on the GitLab dashboard, where the top navigation bar exposes a To-Do List entry. Two candidate actions target the same element [64]: Action Candidate 1 performs click and Action Candidate 2 performs hover, as shown in Fig. 8.

WebArbiter induces four principles, assigning the highest weight to Correctness & safety (40%). In its structured justification, the model rates both candidates as correct in element reference, but judges that hovering "preserves flexibility while gathering information without committing to navigation," while clicking "may trigger unnecessary navigation." The verdict therefore favors Action Candidate 2 (hover). However, the accessibility tree does not encode whether hovering over a given element triggers any interactive response (e.g., a tooltip or dropdown). In this specific page context, hover produces no state change and only introduces a redundant step. Because the actual interaction effect of hover is absent from the text-only observation, the model overestimates its value under the safety principle. This pattern illustrates a shared challenge for text-based WebPRMs: evaluating the execution-level consequence of an interaction type requires information beyond what accessibility trees provide, pointing to visual observations or environment feedback as a promising direction for future work.

### J.2  ELEMENT REFERENCE HALLUCINATION

The task is to set the user's GitLab status to "Enjoying life." Two candidate actions are considered: Action Candidate 1 performs click [6780] with the thought "The Set status option allows me to update my current GitLab status," and Action Candidate 2 performs click [6770] with the thought "Opening Edit profile should provide access to personal settings, including status configuration," as shown in Fig. 9. Crucially, the bid [6780] referenced by Action Candidate 1 is incorrect; the actual "Set status" element corresponds to [6678].

WebArbiter induces "Correct element reference (25%)" as one of its principles, indicating that the model recognizes the importance of verifying bid validity. In the structured justification, the model evaluates both candidates' element references as correct, noting "click [6780], a clickable menu option" and "click [6770], a clickable menu option." It then assigns higher preference to Action Candidate 1 based on its superior task alignment (60% weight). The core issue is that the candidate's thought description carries strong semantic alignment with the task instruction, and the model relies on this alignment rather than independently cross-referencing the bid against the accessibility tree. Accessibility trees are inherently difficult to parse for fine-grained element identification, and when a candidate's thought description is highly task-aligned, text-based WebPRMs tend to trust the semantic signal over low-level bid verification. This challenge is not specific to WebArbiter but applies broadly to WebPRMs operating under text-only observations, suggesting that integrating visual observations or explicit element-level verification mechanisms could help mitigate such grounding errors.

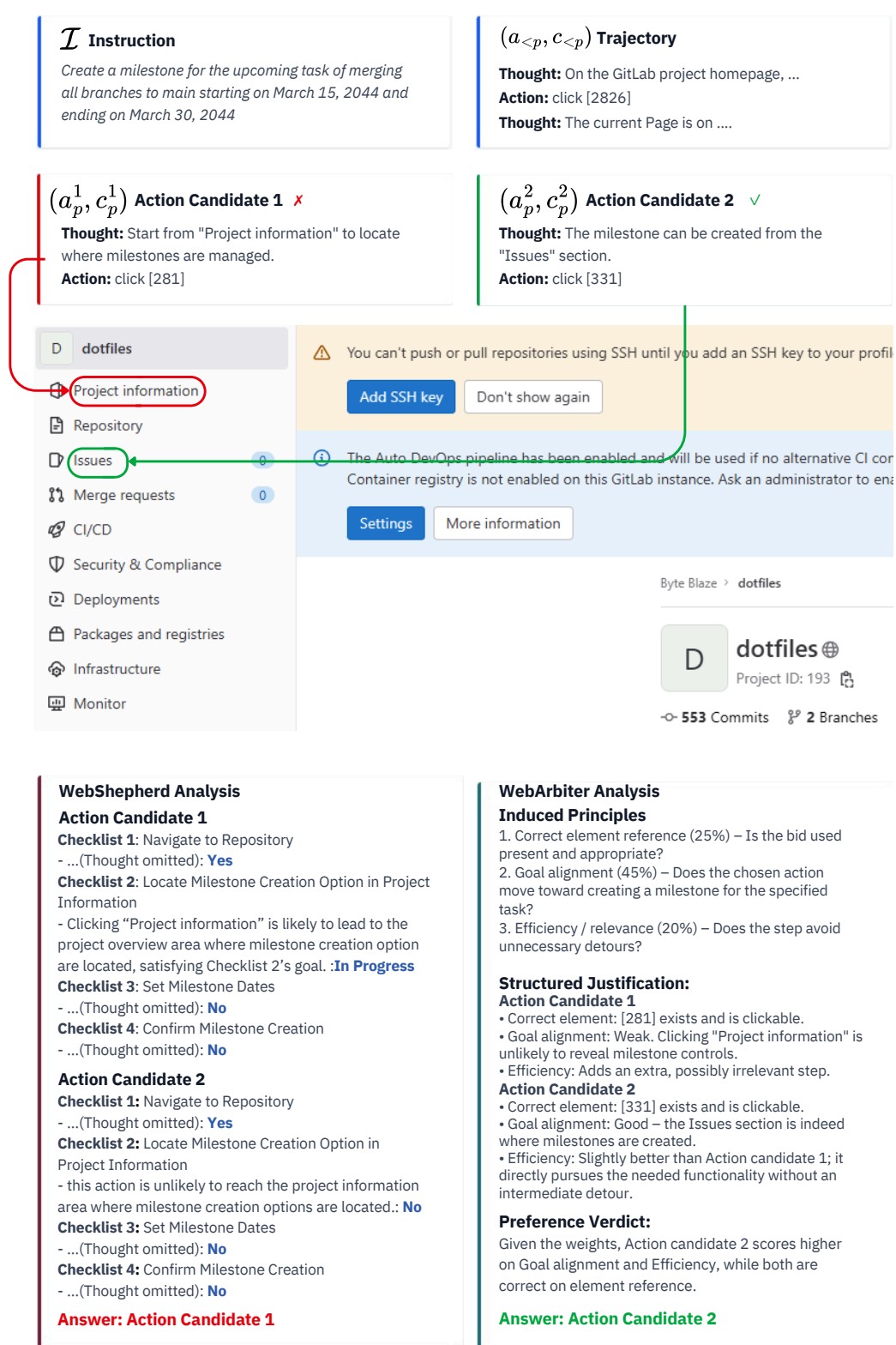

Figure 6: Milestone creation under multiple equivalent paths in GitLab. Checklist-based WebShepherd prefers a procedurally typical but non-essential navigation step under path multiplicity, while WebArbiter reasons over the current state and correctly selects the action that directly advances milestone creation.

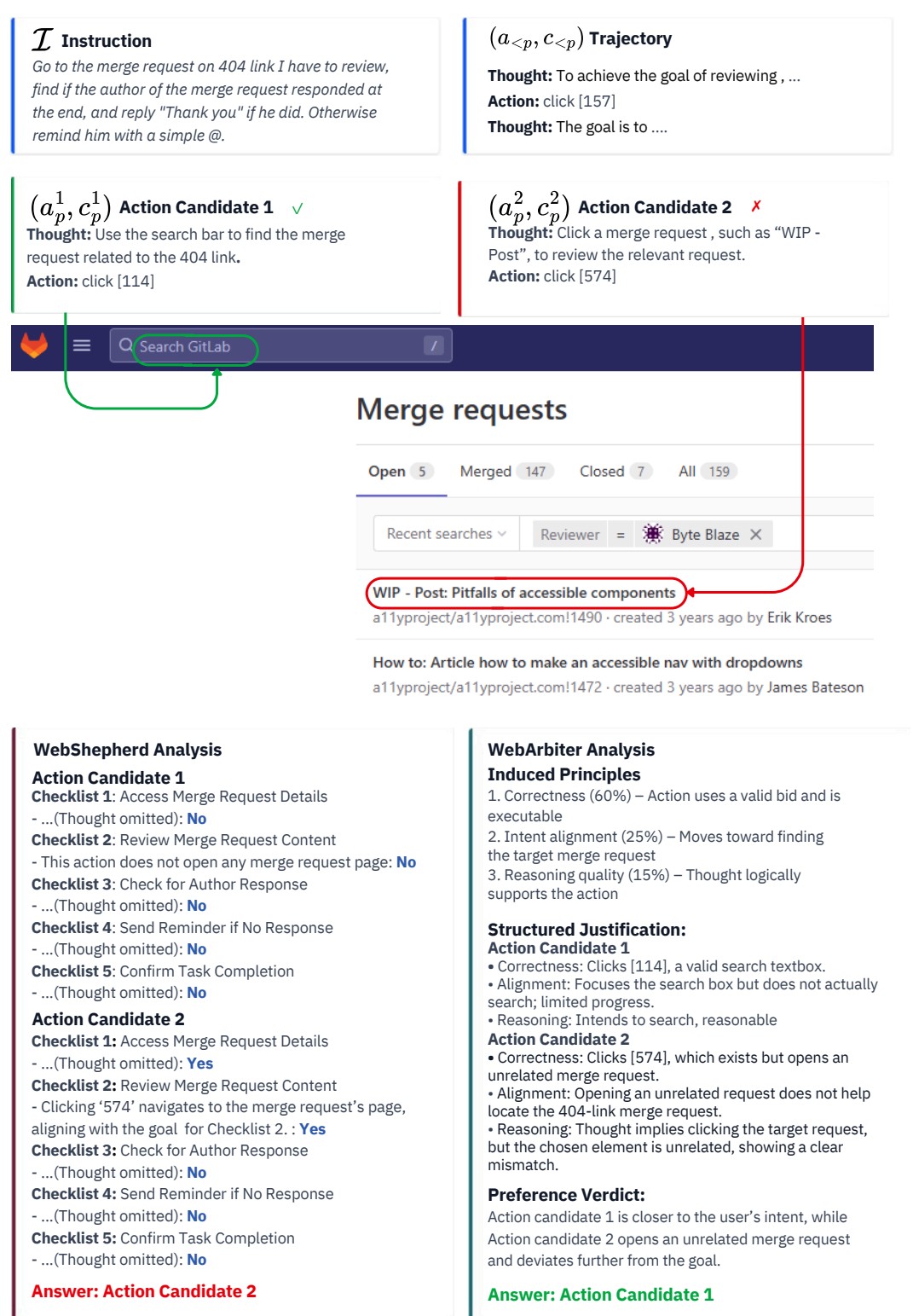

Figure 7: Merge request identification under an ambiguous context. When the target merge request is not yet identified, WebShepherd prematurely commits to an arbitrary request, whereas WebArbiter reasons about task preconditions and prioritizes disambiguation via search.

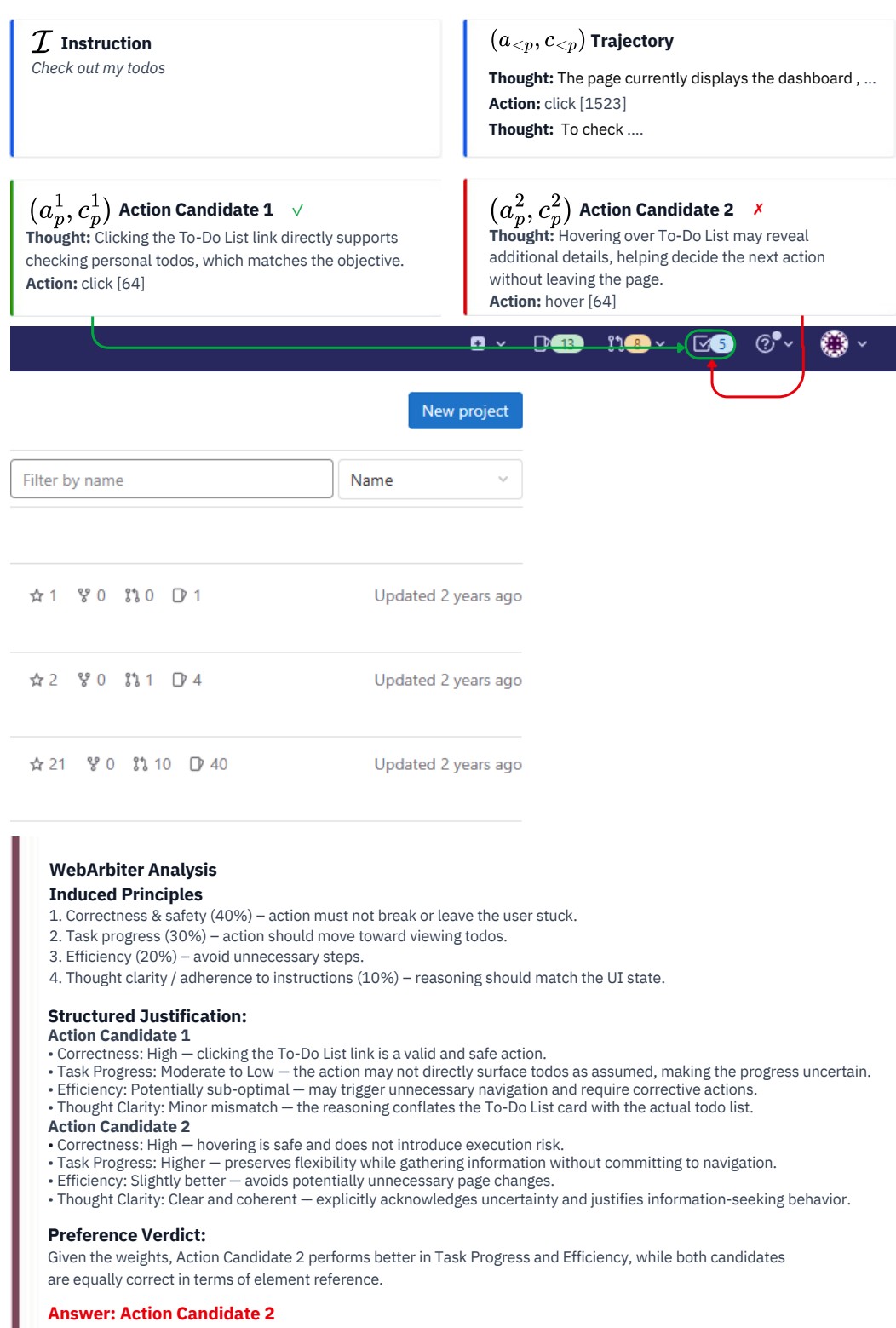

Figure 8: Safe-action bias in reward-guided trajectory search on GitLab. Both candidates target the same To-Do List element, but WebArbiter incorrectly prefers `hover` over `click`. The accessibility tree does not encode the interaction effect of hovering, leading the model to overestimate its value under the safety principle.

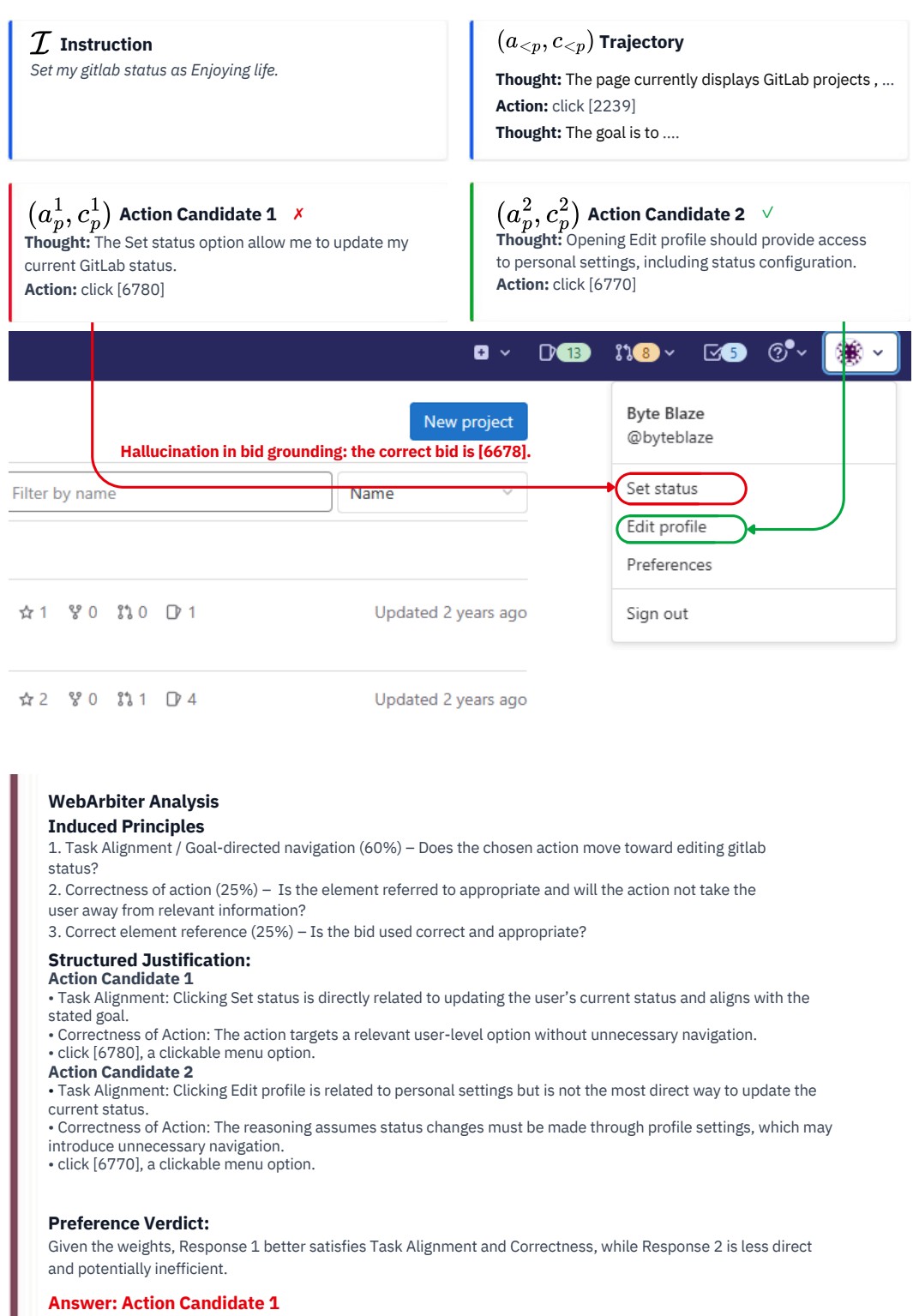

Figure 9: Element reference hallucination in reward-guided trajectory search on GitLab. Action Candidate 1 references an incorrect bid ([6780] instead of the correct [6678]), but WebArbiter prefers it due to its strong semantic alignment with the task. The model induces a "Correct element reference" principle yet fails to detect the bid grounding error from the accessibility tree alone.

