# OpenReview forum: "WebArbiter: A Generative Reasoning Process Reward Model for Web Agents"
_ICLR.cc/2026/Conference — ICLR 2026 Poster_

### Official Review · Reviewer_wacB · 2025-10-27

**Soundness:** 3
**Presentation:** 3
**Contribution:** 2
**Rating:** 6
**Confidence:** 4

**Summary:**

WebArbiter introduces a principle-grounded, generative process reward model designed to evaluate reasoning trajectories in web agents. The model first undergoes reasoning distillation via SFT to induce explicit, interpretable principles from a teacher LLM, and is subsequently optimized through GRPO using binary verifiable rewards. It employs LoRA adapters on top of Qwen2.5-Instruct (3B/7B) for efficient RL training. The authors also propose WEBPRMBench, a new benchmark covering multiple web-agent environments (Mind2Web, WebArena, AssistantBench, WorkArena).

**Strengths:**

1. The PRM is not just a classifier but a generative verifier. It produces explicit reasoning chains before outputting a verdict token. This design enables process-level supervision.
2. Rewards are automatically derived from web-environment logs, whether the predicted “correct” or “incorrect” verdict matches the actual success of an action. This eliminates extra human annotator costs.
3. WebArbiter achieves strong accuracy not just on Mind2Web (its training base) but also on unseen environments like AssistantBench and WorkArena, showing transfer capability. This indicates its principle-based reasoning generalizes beyond template matching.

**Weaknesses:**

1. The binary verifiable reward assumes one correct action per state, which is unrealistic. In tasks in web environment, several actions could succeed, but only one is labeled “Correct.” This introduces false negatives and reward noise,  a core limitation in deterministic PRM setups.
2. The evaluation omits new RL-based WebAgent baselines such as WebAgent-R1 [1] and WebSailor [2] that achieve strong results on WebArena and QA tasks.
3. Using Qwen2.5-7B instead of Qwen3-8B (as in WebShepherd) introduces an architecture-generation gap. Qwen3 includes better instruction alignment, longer context, and multi-turn reasoning improvements. Reported performance gains in WebArbiter may partly reflect architectural differences, not purely training-method superiority.

[1] Wei, Zhepei, et al. "WebAgent-R1: Training Web Agents via End-to-End Multi-Turn Reinforcement Learning." ICML 2025 Workshop on Computer Use Agents.

[2] Li, Kuan, et al. "WebSailor: Navigating Super-human Reasoning for Web Agent." arXiv preprint arXiv:2507.02592 (2025).

**Questions:**

1. Why was Qwen2.5 chosen instead of Qwen3-8B, given that baselines (WebShepherd) use newer backbones?
Would results hold under an architecture-matched setting?

2. Does the WebArbiter generalize to challenging QA tasks that involve multi-round web search, like BrowseComp and HLE?

---

> ### Author Response · Authors · 2025-11-27
>
> Thank you for the constructive feedback. We address each point below and have incorporated the relevant updates into the revised paper.
>
> 1. W1 Binary reward: To clarify, our method does **not** assume that a state has only one uniquely correct action. WebArbiter is trained on pairwise preferences, where the model only determines whether the expert action is preferred over a sampled candidate. If multiple actions could legitimately advance the task, they are simply not used as negative samples; they are not labeled as “incorrect.” The 0/1 reward reflects only whether the preference judgment is correct, not whether an action is uniquely successful. Moreover, the benchmark construction explicitly avoids false negatives. As described in Appx. F, all candidate negative actions undergo rule-based filtering and manual review to discard any alternatives that remain potentially valid. When more than four valid rejected actions remain, we randomly sample a subset to maintain a consistent number of negative candidates per state. This ensures that negative samples reflect true environment-level failures rather than alternative-but-valid execution paths. Finally, WebArbiter’s principle-inducing reasoning relies on structural progress principles, rather than memorizing a single gold label, making the model robust even when multiple valid continuations exist.
>
> 2. W2 Missing RL baselines: WebAgent-R1 and WebSailor focus on end-to-end RL **policy learning**, whereas our work targets WebPRMs, process-level reward models that provide step-wise, multi-candidate preference signals. RL agents do not produce such signals and therefore cannot serve as PRM baselines nor replace a PRM during inference-time search. These two families are complementary rather than competing: a PRM like WebArbiter can act as a reward/critic module for RL agents, while RL policies can benefit from PRM-guided supervision. To isolate the effect of the PRM itself, our evaluation fixes the policy LLM (GPT-4o-mini/4o) and varies only the PRM (no-PRM, self-judge, WebShepherd, and WebArbiter). This setup ensures performance differences primarily reflect PRM quality rather than policy capacity. As shown in Tab. 4, WebArbiter yields the largest gains. Notably, when using GPT-4o as the policy, WebArbiter reaches an average SR of 49.15, higher than WebAgent-R1’s reported 44.8 on the same environments. Since WebSailor focuses on search/visit information-retrieval agent rather than a full web-interaction policy, it does not report results on WebArena.
>
> 3. W3 Qwen2.5 vs Qwen3: We agree with the general expectation that Qwen3 models provide stronger instruction following and reasoning capabilities than Qwen2.5. However, despite using the smaller Qwen2.5-7B base model, WebArbiter already outperforms WebShepherd built on the larger Qwen3-8B, indicating that our gains are not attributable to architecture but to the proposed PRM design and training methodology. We therefore expect an even larger improvement when switching WebArbiter to Qwen3. This experiment is currently in progress: preliminary SFT-only results with Qwen3-8B already surpass WebShepherd’s performance, and GRPO training is underway. We will include the full Qwen3-based results in the final version.
>
> 4. Q1 Why choose Qwen2.5: We selected Qwen2.5 as the backbone because WebShepherd also uses Qwen2.5 for its 3B setting; starting from the same base model enables a clean, architecture-matched comparison at the small-model scale. Since our work begins with small-model PRMs, Qwen2.5–3B provides a controlled and widely used starting point. A Qwen3–8B version of WebArbiter is currently under training.
>
> 5. Q2 Generalization to QA tasks: Our evaluation on ASSISTANTBENCH, one of the four environments included in WebPRMBench, directly targets the setting of challenging QA with multi-round web search: tasks require multi-step browsing across multiple real-world sites, aggregating evidence, and producing structured answers. This setting is closely aligned with the goals of benchmarks such as BrowseComp and HLE. Although we have not run WebArbiter on those specific suites due to time constraints, the strong gains observed on ASSISTANTBENCH indicate that WebArbiter’s process rewards generalize beyond navigation to complex multi-hop web QA. In particular,  WebArbiter-7B achieves a BoN accuracy of 72.41 on ASSISTANTBENCH, substantially higher than WebShepherd-8B’s 44.83 and also exceeding the strongest non-WebPRM baseline, GPT-4o-mini, which attains 68.97.

---

### Official Review · Reviewer_Pe8d · 2025-10-30

**Soundness:** 2
**Presentation:** 2
**Contribution:** 2
**Rating:** 4
**Confidence:** 5

**Summary:**

This paper introduces **WebArbiter**, a novel Process Reward Model (WebPRM) designed to provide step-level supervision for web navigation agents. Unlike traditional outcome-based reward models that offer sparse and delayed feedback, or existing scalar and checklist-based WebPRMs that are brittle and lack interpretability, WebArbiter formulates reward modeling as a text generation task. It produces structured justifications that conclude with a preference verdict, identifying the action most conducive to task completion. The model is trained in two stages: first, reasoning distillation from a stronger teacher LLM to instill principle-guided reasoning; second, reinforcement learning to align verdicts with correctness and improve generalization. The authors also released **WEBPRMBENCH**, a comprehensive benchmark across four diverse web environments (Mind2Web, WebArena, AssistantBench, WorkArena).

**Strengths:**

- **Interpretability and Robustness:** WebArbiter’s reasoning-first approach provides auditable justifications, making it more interpretable and less reliant on superficial cues compared to scalar or checklist-based methods.
- **Effective Training Pipeline:** The two-stage training (reasoning distillation + RL) effectively combines the benefits of principled reasoning and correctness alignment.

**Weaknesses:**

- **Limited Multilingual Support:** The paper does not discuss multilingual capabilities, which limits its applicability in global web environments.
- **Potential for Reward Hacking:** While the RL stage aligns verdicts with correctness, the risk of reward hacking (e.g., over-optimizing for superficial patterns in justifications) is not addressed.
- **Narrow Benchmark Scope:** Although diverse, WEBPRMBENCH is limited to four environments, and its real-world coverage (e.g., multilingual or highly dynamic sites) is unclear.

**Questions:**

1. **Multilingual Generalization:** Does WebArbiter support non-English web environments? If not, are there plans to extend its reasoning and principle induction to multilingual contexts?
2. **Reward Hacking Mitigation:** How does the model avoid reward hacking during RL training? Are there mechanisms to ensure that the generated justifications genuinely reflect task progress rather than exploiting shortcuts?
3. **Scalability and Efficiency:** What are the inference latency and computational requirements of WebArbiter compared to simpler WebPRMs? How feasible is it for real-time web agent control?

---

> ### Author Response · Authors · 2025-11-27
> **Comment on Reviewer Weaknesses**
>
> Thank you for the constructive feedback. We address each point below and have incorporated the relevant updates into the revised paper.
>
> 1. W1 Limited Multilingual Support: Current WebAgent benchmarks (Mind2Web, WebArena, AssistantBench, WorkArena) are all English-only, and no existing WebPRM or WebAgent work provides explicit multilingual capability. Thus, this limitation reflects a gap in the available environments rather than a constraint of our approach. Importantly, WebArbiter’s decision-making relies primarily on state, executability cues, and state-transition consistency, rather than language-specific semantics, making the model naturally extensible to multilingual webpages. While multilinguality has not yet been a primary focus of the WebAgent community, it is indeed important, and we view multilingual WebPRMs as a promising direction that we are excited to explore in future work.
> 2. W2 Potential for Reward Hacking: WebArbiter incorporates two structural defenses that substantially reduce this risk. First, the RL reward is purely verifiable: it is computed only from whether the model’s verdict matches the environment-grounded preferred action (0/1 correctness), not from the justification text. Thus, generating superficially plausible Criteria/Analysis offers no benefit, incorrect actions always yield zero reward. Second, the principle-inducing reasoning format requires the model to ground its judgment in executable progress signals, e.g., action viability, forcing reasoning to be anchored in environment dynamics rather than textual patterns. Together, these mechanisms make the model resistant to reward hacking driven by justification heuristics.
> 3. W3 Narrow Benchmark Scope: While the benchmark currently contains four environments, these are substantially richer and more diverse than the existing publicly available WebPRM benchmark. The only existing benchmark (WebShepherd’s WebRewardBench) includes just two environments with limited WebArena coverage. In contrast, WebPRMBench expands to Mind2Web, WebArena, AssistantBench, and WorkArena, jointly covering consumer-facing and enterprise tasks, real-world websites (shopping, CMS, reddit, GitLab), and open-world multi-step workflows (IT/HR). This provides broad coverage of heterogeneous, dynamic real-world sites. Regarding multilinguality, the absence of multilingual evaluation reflects a limitation of current public WebAgent environments rather than of our method. Since our PRM relies on raw state, executability signals, and trajectory grounding, rather than language-specific semantics, it readily extends to multilingual webpages. Expanding WebPRMBench to multilingual sites is a natural next step and part of our planned future work.

---

> ### Author Response · Authors · 2025-11-27
> **Comment on Reviewer Questions**
>
> 1. Q1 Multilingual Generalization: Thank you for the question. Our method is inherently language-agnostic: its decision process relies primarily on current state, executability signals, and trajectory-level grounding, rather than language-specific semantics. Since the underlying base models we use are multilingual, extending the method to multilingual webpages is straightforward. The current benchmark does not include multilingual evaluation simply because no publicly available multilingual WebAgent environments exist. Extending both our benchmark and evaluation to multilingual settings is an important and natural direction, and we plan to pursue this in future work.
> 2. Q2 Reward Hacking Mitigation: WebArbiter is explicitly designed to prevent reward hacking by decoupling reward signals from the justification text and grounding them in verifiable environment outcomes. As detailed in our RL stage (§3.3.3), the model receives a binary reward solely based on whether the final verdict matches the ground-truth preferred action. The justification content is never used as a reward proxy. Therefore, producing superficially well-written Criteria/Analysis that do not correspond to correct task progress yields a reward of 0 and triggers a negative GRPO update. This aligns the optimization target with genuine task progress rather than linguistic fluency. Moreover, because Web navigation is a partial-observability, state-dependent POMDP (§3.1), the preferred action often depends on subtle executability constraints and observation. These constraints cannot be exploited by textual shortcuts: only reasoning chains that correctly anchor to state + history lead the model toward the action that truly advances the task. Finally, our principle-inducing structured reasoning (§3.3.2) creates an additional barrier against reward hacking. Principles such as task progress and state correctness must be instantiated from the current observation, and the subsequent Analysis stage evaluates candidate actions under these principles. Because the final reward is governed entirely by the correctness of the verdict, the justification serves as a structural scaffold rather than an optimization target. This design makes reward hacking via superficial explanation patterns highly ineffective in practice.
> 3. Q3 Scalability and Efficiency: As reported, the 3B/7B versions of WebArbiter have similar per-call latency to existing WebPRMs such as WebShepherd-3B/8B and other 3B–8B reasoning RMs; the main overhead comes from producing a structured rationale. At the same time, WebArbiter remains much lighter than large self-reflective LLMs, e.g., GPT-5, so it does not introduce an additional order of magnitude in compute. For real-time control, PRM evaluation is lightweight: it only needs to produce a structured rationale and a binary verdict, which is substantially cheaper than generating a full action step. In practice, PRM inference is faster than a single policy, e.g., GPT-4o, action generation, so the additional latency introduced by WebArbiter is negligible relative to the policy LLM. Therefore, WebArbiter does not bottleneck real-time agent control. In settings that require stricter latency, WebArbiter can be queried sparsely (e.g., only on branching or high-risk states) or batched across parallel trajectories, making its use in online control feasible while preserving the benefits of process-level reasoning.
> | Model | Latency (s) |  |
> | --- | --- | --- |
> | R3-Qwen3-4B-LoRA-4k | 4.2907 |  |
> | Binary-Think-RM-3B | 3.8985 |  |
> | WebShepherd-3B | 3.4598 |  |
> | **WebArbiter-3B** | 3.9275 |  |
> | R3-Qwen3-8B-LoRA-4k | 6.4075 |  |
> | RM-R1-Qwen2.5-Instruct-7B | 5.8722 |  |
> | RRM-7B | 5.3155 |  |
> | Binary-Think-RM-8B | 6.3048 |  |
> | WebShepherd-8B | 6.2629 |  |
> | **WebArbiter-7B** | 5.8909 |  |
> | **GPT-OSS-20B** | 10.2078 |  |
> | GPT-4o-mini | 8.5044 |  |
> | GPT-4o | 9.1718 |  |
> | GPT-5 | 44.9520 |  |
> | Claude-sonnet | 11.1658 |  |
> | Gemini-Flash | 3.2283 |  |
> | DeepSeek R1 | 31.2035 |  |

---

### Official Review · Reviewer_EzxW · 2025-10-31

**Soundness:** 2
**Presentation:** 3
**Contribution:** 2
**Rating:** 4
**Confidence:** 3

**Summary:**

This paper proposes WebArbiter, a principle-guided reasoning process reward model (PRM) for web agents. It aims to improve the interpretability, reliability, and robustness of reward modeling for long-horizon web interaction tasks. Unlike scalar WebPRMs (which output coarse numeric rewards) or checklist-based generative WebPRMs (which rely on fragile templates), WebArbiter formulates reward modeling as structured text generation—producing reasoning chains that derive principles, compare candidate actions, and conclude with a preference verdict. The model is trained via a two-stage pipeline:
(1) Reasoning distillation from a stronger teacher LLM to teach principle-based justification, and
(2) Reinforcement learning with verifiable correctness rewards to align reasoning with factual outcomes.
Experiments on the newly released WEBPRMBENCH benchmark and WebArena-Lite show that WebArbiter achieves significant performance improvements over previous WebPRMs and even surpasses strong LLM-as-judge baselines like Gemini Flash and GPT-5.

**Strengths:**

1.	Solid empirical improvement: WebArbiter-7B achieves +10.9% BoN accuracy over Gemini Flash and +48% over WebShepherd, showing consistent advantages across diverse benchmarks.
	2.	Interpretability: The model generates explicit, auditable reasoning chains for each action decision.
	3.	Benchmark release: The introduction of WEBPRMBENCH enriches the evaluation ecosystem for process reward models in web agents.
	4.	Thorough ablation: The authors analyze the roles of principle guidance, reasoning distillation, and RL alignment in model performance.

**Weaknesses:**

1.	Limited conceptual novelty —
The paper’s design (reasoning-based PRM + two-stage pipeline) closely mirrors prior reasoning reward models like RM-R1, RewardReasoningModel, and Generative Verifier. The only difference lies in applying it to the web navigation domain. The overall framework lacks a strong theoretical or methodological innovation beyond scene adaptation.
2.	Missing key baselines —
Table 2 compares WebArbiter only against WebShepherd and a few LLM-as-judge models. However, it omits recent strong generative reward models (e.g., Rubric-RM, Generative Verifier from DeepSeek), which are more directly comparable. This omission weakens the evidence for claimed superiority.
3.	No evaluation of downstream supervision quality —
In Table 4, the paper reports WebArbiter’s gains in reward-guided trajectory search but does not compare against the downstream effects of other PRM or generative models listed in Table 2. Without showing whether the higher WEBPRMBENCH score translates into better supervised model performance, it’s hard to confirm the practical utility of the learned reward model.
4.	Lack of comparison with reasoning-capable LLMs —
Table 4 focuses on reward-guided trajectory search with GPT-4o-mini and GPT-4o, but omits comparisons with stronger reasoning models such as GPT-5 or DeepSeek-R1, which already exhibit built-in search and reflection abilities. These models might achieve comparable or better results without explicit PRM supervision, calling into question whether WebArbiter’s explicit reward modeling still offers clear advantages.
5.	Benchmark dependency —
WEBPRMBENCH is designed and used solely by the authors, which may limit reproducibility and objectivity. Its annotation criteria and preference construction may bias evaluation toward the proposed model’s reasoning style.

**Questions:**

1.	Why are DeepSeek-R1 and Rubrics-based reward models not included in the baseline comparison, given their conceptual proximity?
	2.	Have the authors compared downstream supervised agent performance (e.g., post-training agents) when using WebArbiter versus other PRMs?
	3.	How does WebArbiter’s inference latency or search overhead compare to baseline self-reflective reasoning LLMs like GPT-5?
	4.	Can the principle-guided reasoning process generalize to non-web reasoning tasks, or is it specifically tailored to the web environment?
	5.	How is the benchmark annotation quality ensured? Were multiple annotators or verifiers used to avoid label bias?

---

> ### Author Response · Authors · 2025-11-27
> **Response to Weaknesses 1–3**
>
> Thank you for the constructive feedback. We address each point below and have incorporated the relevant updates into the revised paper.
>
> **W1 Limited conceptual novelty:** We appreciate the reviewer’s comment and would like to clarify a key conceptual distinction. The cited prior works, e.g., RM-R1, are all Outcome Reward Models (ORMs) that score static text responses without access to environment state, action history, or executability. They operate purely on surface-form text and were never designed for multi-step, stateful environments. In contrast, web agents require Process Reward Models (PRMs): web navigation involves long horizons, partial observability, and irreversible actions, where outcome-based signals are sparse, delayed, and often ambiguous. Reliable agent learning demands dense, step-level, state-grounded supervision, which ORMs cannot provide and thus cannot support e.g., inference-time scaling.
>
> Moreover, WebPRM introduces challenges not studied in prior reasoning-based RMs. A WebPRM must determine whether each candidate action genuinely advances task progress under dynamic states, shifting layouts, and partial observations. This includes: (1) the state-grounding challenge, i.e., tying judgments to the actual browser state rather than textual heuristics; (2) the action executability and progress-verification challenge, since plausible-looking text descriptions often do not meaningfully change the environment; (3) robustness to layout and semantic drift, which commonly break checklist- or template-dependent methods; and (4) avoiding spurious linguistic correlations, where fluent reasoning can mask incorrect actions. As shown in the revised paper (Appx. G), when directly applied to Web tasks, existing reasoning-based RMs perform poorly on WebPRMBench: they often reward actions that appear reasonable in text but do not advance the task, cannot determine whether an action is executable from the current observation, and are easily misled by superficial textual plausibility.
>
> WebArbiter directly addresses these challenges, with each component validated by our ablations in §5.1.3. It grounds evaluations in accessibility-tree observations and action history, ensuring judgments reflect the actual browser state. It applies principle-inducing reasoning to derive state- and task-specific criteria that verify whether an action genuinely advances progress.
>
> **W2 Missing key baselines:** Following the reviewer’s suggestion, we have added recent publicly available generative reward models to Tab. 5 and evaluated them on WebPRMBench. As discussed above, these models exhibit limited transferability to procedural, state-dependent Web tasks: they frequently reward actions that appear textually plausible yet fail to advance the task, lack the ability to assess executability from the current observation, and are easily misled by superficial linguistic cues. This is consistent with our finding that outcome-based or verbal-scoring RMs lack the state grounding and progress verification required for WebPRMs. The Generative Verifier is not publicly released, preventing evaluation under our standardized protocol; we now clarify this in the paper and include additional analyses in Appx. G. Notably, our WebArbiter-7B substantially outperforms the strongest generative RM baseline (RRM-7B), achieving an Avg. BoN Accuracy of 77.78% versus 55.32%, a +22.46 absolute improvement across environments.
>
> **W3 No evaluation of downstream supervision quality:** Due to time constraints, we were unable to run a full downstream reward-guided search for all baselines. Instead, consistent with our goal of assessing process-level supervision quality, we apply the reward-guided trajectory search setup in WebArena-Lite (§5.2) to the top-performing models on WebPRMBench, using GPT-4o-mini as the fixed policy. As illustrated in the table below, models with higher WebPRMBench scores exhibit a clear positive correlation with higher average SR under search, indicating that stronger step-level PRMs translate into more reliable multi-step action selection. WebArbiter yields the largest improvements, suggesting that its principle-inducing, state-grounded reasoning provides more effective downstream guidance for long-horizon web interactions.
> |  | Reward-guided Trajectory Search (Avg SR) | WebPRMBench (Avg BoN Acc) | WebPRMBench (Avg Pairwise Acc) |
> | --- | :---: | :---: | :---: |
> | Gemini | 29.46 | 66.93 | 85.70 |
> | Claude | 31.78 | 62.05 | 81.89 |
> | GPT-5 | 27.91 | 60.42 | 79.12 |
> | GPT-4o-mini | 26.74 | 60.70 | 83.83 |
> | WebArbiter-7B | 41.04 | 77.78 | 87.08 |

---

> ### Author Response · Authors · 2025-11-27
> **Response to Weaknesses 4-5 and Questions**
>
> **W4 Lack of comparison with reasoning-capable LLMs:** Thank you for the question. We agree that strong reasoning-capable LLMs are valuable points of comparison. However, recent results from the WebAgent-R1 paper indicate that reasoning LLMs alone do not solve web control reliably: even advanced models, e.g., OpenAI-o3, achieve moderate SR, far below the level required for consistent long-horizon web navigation. Their built-in search and reflection abilities do not translate into stable step-level credit assignment or state-grounded procedural correctness. This directly supports our motivation: reasoning ability alone is insufficient, and explicit process-level reward modeling remains necessary. WebArbiter is designed to provide state-aware, action-level verification, which complements rather than duplicates LLM reasoning. Moreover, our reward-guided trajectory search experiments already demonstrate that stronger PRMs yield better downstream behavior even when paired with powerful LLMs.
>
> | Method | Reddit | GitLab | CMS | Shopping | Avg |
> | --- | :---: | :---: | :---: | :---: | :---: |
> | QwQ-32B  | 15.8 | 33.3 | 25.7 | 20.0 | 23.7  |
> | OpenAI-o3 | 36.8 | 46.7 | 45.7 | 33.3 | 40.6  |
> | OpenAI-o4-mini | 47.4  | 43.3 | 45.7 | 28.9 | 41.3 |
> | GPT-4o + WebArbiter | 44.44 | 42.86 | 52.63 | 56.67 |  49.15 |
>
> **W5 Benchmark dependency:** We appreciate the reviewer’s concern. WebPRMBench fills a clear gap in existing resources. The only publicly available WebPRM benchmark (WebRewardBench from WebShepherd) covers just two environments with limited WebArena coverage. In contrast, WebPRMBench spans four substantially richer and more diverse environments, covering consumer and enterprise workflows, real-world websites, and multi-step open-ended tasks. Importantly, WebPRMBench is explicitly constructed to avoid bias toward any particular reasoning style. As detailed in Appx. F, positive samples come from human-validated minimal-step trajectories, where annotators verify monotonic progress and revise deviations to obtain the shortest successful path. For negatives, alternative actions are drawn from diverse policy models and filtered purely by environment-grounded executability and progress, not by linguistic form. Filtered candidates are manually reviewed, and when more than four valid rejected actions remain, a random subset is selected to maintain consistency. These steps ensure that preferences reflect actual environment dynamics, not the proposed model’s justification style. WebPRMBench is intended as a reusable resource for the community, and we will release the benchmark to support further research.
>
> **Q1 Why no results from reasoning-based RMs:** We have added these results in the revision (Appx. G). As shown, WebArbiter substantially outperforms these reasoning-based RMs and DeepSeek-R1 on WebPRMBench.
>
> **Q2 Comparison with PRM-supervised downstream agents:** At present, no existing web agent is trained with PRM-based supervision, including WebAgent-R1, which relies on binary outcome rewards rather than model-based, step-level rewards. Therefore, there is no supervised post-training agent baseline that uses PRMs for us to compare against. A full PRM-supervised post-training pipeline is beyond the scope of this paper and will be explored as future work.
>
> **Q3: Inference latency:** As shown below, WebArbiter’s inference latency is comparable to other 7B–8B reasoning reward models. At the same time, it is substantially more efficient than large self-reflective LLMs, e.g., GPT-5, which incur significantly higher computational overhead.
>
> | Model | Latency (s) |  |
> | --- | --- | --- |
> | R3-Qwen3-8B | 6.4075 |  |
> | RRM-7B | 5.3155 |  |
> | Binary-Think-RM-8B | 6.3048 |  |
> | RM-R1-Qwen2.5-Instruct-7B | 5.8722 |  |
> | WebShepherd-8B | 6.2629 |  |
> | **WebArbiter-7B** | 5.8909 |  |
> | GPT-OSS-20B | 10.2078 |  |
> | GPT-5 | 44.9520 |  |
>
> **Q4: Generalization to non-web reasoning tasks:** The principle-guided reasoning mechanism is not specific to the web domain. Its core idea, deriving progress principles from state transitions, grounding them in observation, and applying them to compare candidate actions, is domain-agnostic. WebArbiter instantiates this framework using web-specific signals such as DOM/AxTree and executability cues, which naturally arise in web environments. More broadly, the same mechanism applies to settings such as interactive GUIs, tool-use and API-based agents, embodied control, and other procedural or stateful reasoning tasks, etc. Extending principle induction beyond the web setting is a natural direction for future work.
>
> **Q5: Benchmark annotation quality:** All labels come from environment-grounded verification rather than annotator preference, guaranteeing that preference pairs reflect actual environment outcomes rather than subjective interpretation. Negative actions are sampled from diverse policy models and manually reviewed to avoid false negatives. Additional details are provided in Appx. F.

---

### Official Review · Reviewer_xtyG · 2025-10-31

**Soundness:** 3
**Presentation:** 3
**Contribution:** 3
**Rating:** 6
**Confidence:** 2

**Summary:**

The paper presents a novel approach to enhancing the performance of web agents in complex, multi-step tasks. It identifies the limitations of existing reward models, which often rely on coarse scoring or checklist-based methods that lack interpretability and robustness. To address these challenges, the authors introduce WebArbiter, a reasoning-first model that formulates reward modeling as a text generation task. The results demonstrate that WebArbiter significantly outperforms existing models, showcasing its robustness and practical value in real-world web tasks.

**Strengths:**

1. The paper introduces a fresh perspective on reward modeling by framing it as a text generation task. This innovative approach allows for the generation of structured justifications for actions, which is a significant departure from traditional scalar scoring and checklist-based methods. By integrating reasoning distillation and reinforcement learning, the authors create a model that not only enhances decision-making but also provides a more interpretable framework for understanding agent behavior.

2. By addressing the limitations of existing reward models and providing a more reliable framework for web agents, the paper has the potential to impact various applications in automated web tasks. The advancements presented in WebArbiter could lead to more effective and interpretable web agents, ultimately enhancing their utility in real-world scenarios.

**Weaknesses:**

1. The description of the WEBPRMBENCH construction is somewhat abstract. It would be helpful to include concrete examples or illustrations from the benchmark to make the process easier for readers to understand.

2. The main paper references an appendix (e.g., “Appendix XXX”), but no appendix is actually included.

3. The figures in the paper are low-resolution. Please replace them with higher-quality versions to improve readability.

**Questions:**

See above

---

> ### Author Response · Authors · 2025-11-26
>
> Thank you for the constructive feedback. We appreciate the reviewer’s suggestions and have updated the paper accordingly.
>
> 1. Benchmark examples. Examples from WEBPRMBENCH are provided in Appx. B. We have also added a more detailed construction description in Appx. F, and the benchmark will be publicly released. As summarized in Appx. F, WEBPRMBENCH is constructed from successful, expert-verified trajectories in AgentRewardBench, which provides environment-grounded annotations of execution quality. We select only minimal-step trajectories and have annotators validate monotonic progress, remove detours, and revise any deviations to obtain the shortest successful path. Missing rationales are completed to ensure consistent state–action explanations, and the resulting actions, empirically verified to execute successfully in the real environment, serve as positive labels. Negative actions are obtained by sampling diverse policy models and applying rule-based filters to remove alternatives that might still succeed; only invalid or non-progressing actions are kept to reflect true environment-level failures. To ensure consistency and avoid false negatives, filtered candidates are manually reviewed, and if more than four valid rejected actions remain after filtering, we randomly sample a subset to maintain a uniform number of negative alternatives per instance.
>
> 2.	Appendix reference. The referenced appendix was previously included in the Supplementary Material; for easier reading, we have now placed it at the end of the main PDF after the references.
>
> 3.	Figure quality. Main figures have been replaced with higher-resolution versions.

---

### Meta-Review · Area_Chair_bHUz · 2026-01-07

**Summary:**

Several reviewers (EzxW, wacB) initially noted the omission of recent generative reward models and reasoning-capable LLMs like DeepSeek-R1 or OpenAI-o3 as baselines. They questioned whether WebArbiter's performance gains were truly due to its methodology or if existing high-capacity models could perform just as well without explicit process reward modeling. This was addressed later during the discussion phase by empirically showing that advantage of the proposed method over these off-the-shelf model.

Overall, I think this is a well-executed application paper that is well-received by the reviewers.

**Reviewer Concerns:**

Addressed:
- Reviewers initially noted the omission of recent generative reward models and reasoning-capable LLMs like DeepSeek-R1 or OpenAI-o3 as baselines. This was addressed later during the discussion phase by adding empirical comparison.
- Reviewer EzxW raised a concern about limited conceptual novelty, suggesting the two-stage pipeline closely mirrored prior reasoning reward models like RM-R1.The authors clarified a fundamental distinction: prior models are Outcome Reward Models (ORMs) operating on static text, whereas WebArbiter is a Process Reward Model (PRM) specifically designed for stateful, partially observable web environments. It addresses web-specific challenges such as action executability, layout drift, and state grounding via the accessibility tree (AxTree), which are not addressed by general-domain RMs.
- Regarding reward hacking, the authors clarified that WebArbiter's RL rewards are binary and verifiable, based solely on the final verdict matching the ground truth, not the justification text. This prevents the model from gaining rewards through superficial but well-written explanations.
- Reviewer wacB questioned the assumption of a single correct action per state and whether the use of the Qwen2.5 backbone created an unfair architectural gap. The authors confirmed that WebArbiter outperforms baselines even with the smaller 7B backbone, suggesting method-driven superiority rather than architectural bias.

Outstanding:
- Reviewer Pe8d raised concerns and evaluation regarding the model's multilingual capabilities, which the authors also agreed.
- Reviewer EzxW questioned whether the reward model could be used to actually train a better agent (post-training) rather than just guiding search at inference time. The authors clarified that no current web agents are trained with PRM-based supervision, and a "full PRM-supervised post-training pipeline" was deemed beyond the scope of the paper to be explored in the future.

**Reviewer Scores:**

Reviewer xtyG might have increased from 6 to 7 as most questions are answered.
Reviewer EzxW indicates that they will update the rating (potentially from 4 to 5/6)
Reviewer Pe8d might have increased from 4 to 6 as most questions are answered.
Reviewer wacB would have kept the rating at 6.

---

### Decision · Program_Chairs · 2026-01-26

Accept (Poster)